# KUROMI: LEARNING WITHOUT DATA-SPACE AUGMENTATION VIA ENERGY-BASED SEMI-SUPERVISED KURAMOTO NEURONS

## ABSTRACT

Semi-supervised learning (SSL) often relies on extensive data augmentation or complex teacher-student structures to generate reliable supervision from unlabeled data. In this work, we propose **Kuromi**, a novel SSL framework that discards data-space augmentation entirely and instead leverages energy-based dynamics for representation learning and pseudo-label generation. Central to **Kuromi** is the *Artificial Kuramoto Oscillatory Neuron* (AKOrN), a biologically inspired dynamic neuron model that encodes inputs as synchronized oscillatory states on a hypersphere. Through unsupervised pre-training, fine-tuning on limited labels, and energy-guided self-training, **Kuromi** produces low-energy, structure-aware feature representations without requiring external regularization. To better exploit labeled data and pseudo-labels, we also propose **Energize**, an iterative, augmentation-free latent-state smoothing method inspired by *Weisfeiler-Lehman* aggregation, which operates entirely in the prediction space without hyperparameters. Extensive experiments on CIFAR-10, SVHN, STL-10, and ImageNet show that **Kuromi** achieves state-of-the-art performance among non-transformer SSL methods and remains competitive with recent transformer-based approaches. Notably, **Kuromi** not only achieves state-of-the-art results on CIFAR-10 and STL-10, but also leads in efficiency with just 12M parameters, 0.18 GFLOPs, and the highest throughput across all baselines.

## 1 INTRODUCTION

Modern learning scenarios are often defined by an abundance of unlabeled data and a scarcity of reliable annotations. Semi-supervised learning (SSL) has emerged as a principled approach to exploit this asymmetry, enabling effective training with minimal supervision. To bridge the supervision gap, most state-of-the-art SSL frameworks rely heavily on data augmentation strategies (Bachman et al., 2014; Sajjadi et al., 2016; Laine & Aila, 2016), ensemble teacher-student training (Laine & Aila, 2016), or large pre-trained backbones (Cai et al., 2022; Sohn et al., 2020; Zhang et al., 2021). However, such pipelines are often computationally intensive, augmentation-dependent, and sensitive to hyperparameter choices. More critically, they obscure the connection between the model's internal dynamics and its confidence in predictions, limiting interpretability and generalization under scarce supervision. Aggressive augmentations disrupt the local spatial coherence and token-level structure of inputs, which are essential for learning stable representations in models like ViTs or SSM networks. The resulting loss of structural alignment can degrade both pseudo-label quality and convergence efficiency (Li et al., 2024).

In this work, we revisit a fundamental question: *Can we design a semi-supervised framework that is data-space augmentation-free and capable of learning structured representations via intrinsic dynamics?* Motivated by this question, we propose **Kuromi**, a novel SSL framework that entirely eliminates data augmentation and teacher-student ensembles. **Kuromi** is powered by **Artificial Kuramoto Oscillatory Neurons (AKOrN)** Miyato et al. (2024), a class of dynamic units inspired by synchronization in coupled oscillators Hubel & Wiesel (1962); Somers et al. (1995); Gray et al. (1989). Instead of conventional static activations, AKOrN encodes inputs as evolving trajectories on a hypersphere, driven by energy minimization Miyato et al. (2024). These dynamics naturally yield structured, low-energy representations and pseudo-labels, without any need for augmentation-

based consistency constraints. The name **Kuromi** fuses *Kuro* from Kuramoto Neurons and *mi* from semi-supervised learning.

To further improve pseudo-label quality, we also introduce **Energize**, a novel latent-state refinement mechanism that operates directly in the prediction space. **Energize** draws conceptual inspiration from the Weisfeiler-Lehman (WL) graph aggregation algorithm: it iteratively smooths predictions across multiple stochastic forward passes, each viewed as a noisy "neighbor"—guided by energy-aware agreement weighting Leman & Weisfeiler (1968); Shervashidze et al. (2011). Unlike WL, **Energize** avoids graph construction and operates in a continuous, prediction space without introducing additional hyperparameters. This results in more calibrated, consistent pseudo-labels while maintaining a lightweight and scalable training pipeline.

To the best of our knowledge, this is the first work to develop an SSL framework based on oscillator dynamics and WL-inspired latent smoothing, all within an augmentation-free design. Extensive experiments across CIFAR-10, SVHN, STL-10, and ImageNet demonstrate that **Kuromi** achieves state-of-the-art results among non-transformer SSL models and remains competitive with modern transformer-based methods. Notably, **Kuromi** achieves **superior efficiency** with only **12M parameters**, **0.18 GFLOPs**, and **2200 img/s** throughput, highlighting its potential for deployment in resource-constrained or real-time settings.

## 2 RELATED WORKS

**Modern Deep SSL Algorithms.** Recent advances in semi-supervised learning (SSL) build on two central ideas: pseudo-labeling ((Scudder, 1965; McLachlan, 1975)) and consistency regularization ((Bachman et al., 2014; Sajjadi et al., 2016; Laine & Aila, 2016)). FixMatch ((Sohn et al., 2020)) combines both by enforcing consistency between weakly and strongly augmented views, setting a strong performance baseline in low-label regimes. However, its reliance on fixed confidence thresholds led to further refinements—FlexMatch ((Zhang et al., 2021)), Dash ((Xu et al., 2021)), FreeMatch ((Wang et al., 2022)), and SoftMatch ((Chen et al., 2023))—which employ adaptive calibration to improve robustness.

Despite empirical success ((Sohn et al., 2020; Zhang et al., 2021; Chen et al., 2023)), the theoretical understanding of SSL remains limited. Classical analyses ((Singh et al., 2008; Van Engelen & Hoos, 2020; Wei et al., 2020; Fan et al., 2023; Oliver et al., 2018)) offer generalization bounds under simplified assumptions, while recent studies ((Zhao et al., 2023)) focus on linear models with Gaussian mixtures. More practical SSL designs, such as Semi-ViM ((He et al., 2025)), explore lightweight state-space models for imbalanced settings. Meanwhile, concerns have been raised about augmentation-induced information loss ((Li et al., 2024)), motivating the need for alternative, augmentation-free frameworks.

**Artificial Kuramoto Oscillatory Neurons (AKOrN).** To overcome the limitations of augmentation-based SSL, AKOrN introduces synchronization-based inductive bias inspired by neural computation. Neuroscience studies reveal that lateral interactions in visual cortex lead to synchronized activity and competitive binding ((Hubel & Wiesel, 1962; Somers et al., 1995; Gray et al., 1989; Mountcastle, 1997)), enabling sparse and specialized feature representations ((Amari et al., 1977; Mozer, 1991; Reichert & Serre, 2013; Mysin & Shubina, 2022)). AKOrN models such dynamics using oscillator-based neurons, providing an interpretable and biologically inspired alternative to conventional activation functions.

AKOrN ((Miyato et al., 2024)) models neurons as coupled oscillators governed by a generalized Kuramoto dynamic ((Kuramoto, 2003)), introducing temporal evolution and synchronization into neural computation. Each neuron is a unit vector $\mathbf{x}_i \in \mathbb{R}^N$ constrained on a hypersphere, evolving via:

$$\Delta\mathbf{x}_i = \Omega_i\mathbf{x}_i + \text{Proj}_{\mathbf{x}_i}\left(\mathbf{c}_i + \sum_j J_{ij}\mathbf{x}_j\right), \quad \mathbf{x}_i \leftarrow \text{Normalize}(\mathbf{x}_i + \gamma\Delta\mathbf{x}_i), \quad (1)$$

where $\Omega_i$ denotes intrinsic rotation, $J_{ij}$ is the pairwise coupling strength, and $\mathbf{c}_i$ is an external stimulus. The update enforces alignment both among oscillators and with the external input, minimizing a structured energy functional over time ((Miyato et al., 2024)). In Kuromi, the energy $E$ used during refinement is computed as the total magnitude of these update vectors $\Delta x_i$, so lower energy naturally corresponds to more stable and synchronized oscillator states.

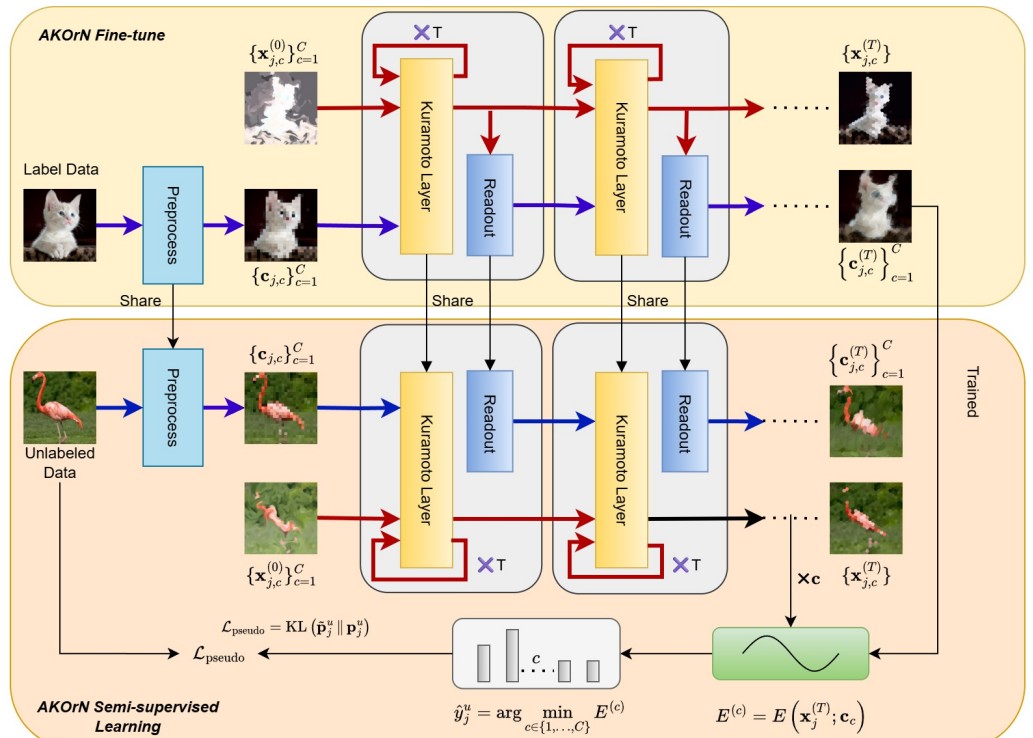

Figure 1: Overall workflow of **Kuromi** for semi-supervised learning. Each input $\mathbf{x}_j^u$ is first encoded into class-wise conditional embeddings $\{\mathbf{c}_{j,c}\}_{c=1}^C$, and oscillator states $\{\mathbf{x}_{j,c}^{(0)}\}_{c=1}^C$ are randomly initialized on a hypersphere. **Red path:** Oscillator states evolve across $T$ Kuramoto steps, resulting in final states $\{\mathbf{x}_{j,c}^{(T)}\}_{c=1}^C$, with projection back to the hypersphere after each step. **Blue path:** Conditional embeddings are propagated through a shared transformation module. Final oscillator states are aggregated via a readout to produce class predictions. Labeled samples use cross-entropy loss, while unlabeled ones are trained via stochastic pseudo-label refinement that minimizes Kuramoto energy $E_j^{(k)}$ computed from $K$ stochastic forward passes indexed by $k$ (see Eq. 7).

AKOrN dynamics offer biologically inspired inductive biases, mirroring cortical synchronization and competition mechanisms observed in neuroscience ((Rubino et al., 2006; Lubenov & Siapas, 2009; Fell & Axmacher, 2011; Zhang et al., 2018; Roberts et al., 2019; Muller et al., 2014; Davis et al., 2020; Zanos et al., 2015)). Unlike static activations, the evolving states naturally produce structured, energy-consistent representations. Moreover, Kuramoto oscillators have been shown to approximate combinatorial optimization ((Strogatz et al., 1989; Wang & Roychowdhury, 2017)), enabling AKOrN to encode reasoning-like behaviors absent in standard deep networks ((Dziri et al., 2023; Zhang & Sheng, 2024)).

**Kuromi** leverages AKOrN as a dynamic backbone for SSL. In contrast to mainstream frameworks like FixMatch ((Sohn et al., 2020)), which rely on handcrafted strong augmentations and confidence thresholds to regularize predictions, Kuromi achieves consistency through intrinsic synchronization and energy minimization. This coordination-driven learning avoids the semantic distortion often induced by aggressive augmentations ((Li et al., 2024)), and offers a principled alternative for training under limited labels without sacrificing generalization.

## 3 PROPOSED FRAMEWORK: *Kuromi*

The overall workflow of **Kuromi** is illustrated in Fig. 1. For each input sample, we denote the set of final oscillator states across all classes as $X = \{\mathbf{x}_{j,c}^{(T)}\}_{c=1}^C$ and the corresponding conditional stimuli as $C = \{\mathbf{c}_{j,c}\}_{c=1}^C$, where $\mathbf{x}_{j,c}^{(T)} \in \mathbb{R}^N$ represents the final state (at Kuramoto step $T$) of the $c$-th oscillator for the $j$-th input, and $\mathbf{c}_{j,c} \in \mathbb{R}^N$ is its class-wise guidance vector. Each oscillator is randomly initialized on a hypersphere, while conditional vectors are data-dependent or class-specific,

guiding synchronization dynamics. The red path in Fig. 1 depicts the evolution of oscillator states through Kuramoto Layers with hypersphere projection, while the blue path propagates conditional embeddings $C$ through a shared transformation module. The stimulus slots remain purely latent dynamical anchors with no semantic correspondence to classes; label information is introduced only through the final classifier.

**Kuromi** operates in three stages. In the *pre-training* stage, the AKOrN backbone is trained on unlabeled data using a contrastive loss to discover low-energy structure via synchronization dynamics. In the *fine-tuning* stage, labeled samples are used to align these dynamics with semantic targets via class-guided supervision and cross-entropy loss. Finally, in the *semi-supervised training* stage, **Kuromi** leverages $K$ stochastic forward passes per unlabeled input to identify low-energy predictions. These are refined into soft pseudo-labels through the proposed **Energize** module, which performs energy-weighted smoothing in functional space. A new forward pass is then trained using KL divergence. Importantly, **Kuromi** eliminates strong data augmentation, relying instead on its intrinsic dynamics for regularization. Each stage is detailed in the following subsections.

## 3.1 AKORN PRE-TRAINING

In this stage, we pre-train the AKOrN backbone using only unlabeled data. The goal is to learn spatially structured, low-energy representations by leveraging the intrinsic synchronization dynamics of Kuramoto oscillators. No label supervision or pseudo-labels are used in this stage.

Given an unlabeled mini-batch $\{\mathbf{x}_j^u\}_{j=1}^{B_u}$, where $\mathbf{x}_j^u \in \mathbb{R}^d$ denotes the $j$-th input image and $B_u$ is the unlabeled batch size, we encode each input into a set of conditional stimulus vectors $\{\mathbf{c}_{j,c}\}_{c=1}^{C}$, where $\mathbf{c}_{j,c} \in \mathbb{R}^d$ corresponds to the $c$-th class-guided conditional embedding for the $j$-th sample.

The $j$-th unlabeled sample is also initialized with a set of oscillator states $\{\mathbf{x}_{j,c}^{(0)}, \mathbf{x}_{j,c}^{(1)}, ..., \mathbf{x}_{j,c}^{(T)}\}_{c=1}^{C}$, where each $\mathbf{x}_{j,c}^{(0)} \in \mathbb{R}^d$ is randomly sampled on a unit hypersphere. These oscillatory states evolve over $T$ discrete Kuramoto steps according to the update rule defined in Eq. 1. After $T$ steps, we obtain the final oscillator states $\{\mathbf{x}_{j,c}^{(T)}\}_{c=1}^{C}$, one for each class $c$, which are subsequently used for prediction and energy computation.

To obtain a global representation for the $j$-th sample, we apply a readout function over the final oscillator states from all $C$ oscillators:

$$\mathbf{z}_j^u = \text{Readout}(\{\mathbf{x}_{j,c}^{(T)}\}_{c=1}^{C}) \in \mathbb{R}^h, \tag{2}$$

where $\mathbf{x}_{j,c}^{(T)} \in \mathbb{R}^N$ is the final state of the $c$-th oscillator for the $j$-th input after $T$ Kuramoto steps, and $\mathbf{z}_j^u$ is the resulting global embedding of dimension $h$.

We train the model using the normalized temperature-scaled cross-entropy (NT-Xent) loss to encourage similar inputs to converge to nearby embeddings while pushing apart dissimilar ones. Let $2B_u$ denote the number of samples after applying two random views to each input (i.e., standard contrastive learning setup). The contrastive loss is defined as:

$$\mathcal{L}_{\text{contrastive}} = \frac{1}{2B_u} \sum_{j=1}^{2B_u} -\log \frac{\exp\left(\text{sim}(\mathbf{z}_j^u, \mathbf{z}_{\text{pos}(j)}^u)/\tau\right)}{\sum_{k=1, k \neq j}^{2B_u} \exp\left(\text{sim}(\mathbf{z}_j^u, \mathbf{z}_k^u)/\tau\right)}, \tag{3}$$

where $\text{sim}(\cdot, \cdot)$ denotes cosine similarity, $\tau$ is a temperature hyperparameter, and $\text{pos}(j)$ returns the index of the positive pair for $\mathbf{z}_j^u$. All other $2B_u - 2$ embeddings are treated as negatives.

This unsupervised pre-training phase encourages the AKOrN backbone to learn robust low-energy representations that reflect structural consistency across randomly initialized dynamics, laying the foundation for subsequent fine-tuning and SSL stages.

## 3.2 AKORN FINE-TUNING

After unsupervised pre-training, we fine-tune the AKOrN backbone using a small labeled dataset to align the learned oscillatory dynamics with semantic supervision. Given a batch of $B$ labeled input samples $\{(\mathbf{x}_i^l, y_i^l)\}_{i=1}^{B}$, where $\mathbf{x}_i^l \in \mathbb{R}^d$ is the $i$-th input image and $y_i^l \in \{1, \ldots, C\}$ is the corresponding ground-truth class label, we follow the same Kuramoto evolution process established in pre-training.

Each labeled sample $\mathbf{x}_i^l$ is combined with a class-specific conditional stimulus vector $\mathbf{C}_{y_i^l} \in \mathbb{R}^d$ to initialize the oscillatory state $\mathbf{x}_i^{(0)}$. The oscillator dynamics evolve over $T$ discrete time steps according to the Kuramoto update rule defined in Eq. 1, resulting in the final state $\mathbf{x}_i^{(T)}$.

We then apply a readout module to map the final oscillator state to a global feature representation:

$$\mathbf{z}_i^l = \text{Readout}(\mathbf{x}_i^{(T)}) \in \mathbb{R}^h, \tag{4}$$

where $\mathbf{z}_i^l$ is the extracted embedding of the $i$-th labeled sample and $h$ is the representation dimension.

This embedding is passed through a classification head consisting of a linear projection followed by softmax to produce the predicted class distribution:

$$\mathbf{p}_i^l = \text{softmax}(W\mathbf{z}_i^l + \mathbf{b}) \in \Delta^C, \tag{5}$$

where $W \in \mathbb{R}^{C \times h}$ and $\mathbf{b} \in \mathbb{R}^C$ are learnable parameters, and $\Delta^C$ denotes the $C$-dimensional probability simplex.

We define the supervised loss as the average cross-entropy between predictions $\mathbf{p}_i^l$ and ground-truth labels $y_i^l$ over the batch:

$$\mathcal{L}_{\text{sup}} = -\frac{1}{B} \sum_{i=1}^{B} \log \left[\mathbf{p}_i^l\right]_{y_i^l}, \tag{6}$$

where $\left[\mathbf{p}_i^l\right]_{y_i^l}$ denotes the predicted probability corresponding to the correct class $y_i^l$.

This fine-tuning stage preserves the dynamic nature of the pre-trained AKOrN model while grounding it with supervised semantic targets. It serves to prepare the model for the subsequent SSL phase, in which both labeled and unlabeled data are incorporated.

### 3.3 AKOrN SSL AND ENERGIZE

We incorporate both labeled and unlabeled data in the semi-supervised training of AKOrN, leveraging its internal energy dynamics to refine pseudo-labels via stochastic oscillation and a novel denoising module called **Energize**.

**Energize** introduces a novel perspective on pseudo-label refinement by shifting from traditional confidence-based thresholding to a functional denoising process grounded in oscillator dynamics. Rather than relying on hand-crafted graphs or hard pseudo-label selection, **Energize** interprets stochastic predictions—generated via multiple Kuramoto evolutions—as soft structural neighbors embedded in a prediction manifold. Inspired by the spirit of the WL algorithm, it re-defines neighborhood aggregation as an energy-weighted alignment process operating entirely in distribution space. This iterative smoothing progressively distills a consensus pseudo-label by favoring low-energy, semantically stable predictions, enabling structure-aware learning without requiring explicit graph construction. Unlike classical WL, which operates over discrete labels and symbolic hashing, **Energize** unfolds as a continuous, differentiable refinement dynamic that inherently preserves prediction uncertainty. The full algorithmic details are presented in Appendix C.

Let the labeled dataset be $\{(\mathbf{x}_i^l, y_i^l)\}_{i=1}^{B_l}$, where $\mathbf{x}_i^l \in \mathbb{R}^d$ and $y_i^l \in \{1, \ldots, C\}$, and the unlabeled dataset be $\{\mathbf{x}_j^u\}_{j=1}^{B_u}$, where $\mathbf{x}_j^u \in \mathbb{R}^d$ and $C$ is the number of classes.

For each labeled input $\mathbf{x}_i^l$, we apply Kuramoto dynamics with class-specific conditional stimulus $\mathbf{C}_{y_i^l}$ to obtain the final state, then apply the readout and classification layers to obtain $\mathbf{p}_i^l \in \Delta^C$. The supervised loss is computed via cross-entropy (as defined in Eq. 6).

For each unlabeled sample $\mathbf{x}_j^u$, we perform $K$ stochastic forward passes with independently initialized oscillator states, obtaining class distributions $\{\mathbf{p}_j^{(k)} \in \Delta^C\}_{k=1}^{K}$ and corresponding energy scores $\{E_j^{(k)}\}_{k=1}^{K}$.

**Energize** begins with the following initialization step:

$$\tilde{\mathbf{p}}_j^{(0)} = \text{Normalize} \left( \sum_{k=1}^{K} \frac{1}{E_j^{(k)}} \cdot \mathbf{p}_j^{(k)} \right), \tag{7}$$

where `Normalize` projects the result back to the probability simplex $\Delta^C$. Then, we perform $T$ iterative refinement steps:

$$\tilde{\mathbf{p}}_j^{(t+1)} = \text{Normalize}\left(\sum_{k=1}^{K} \frac{1}{E_j^{(k)} + \epsilon} \cdot \left(\tilde{\mathbf{p}}_j^{(t)} \odot \mathbf{p}_j^{(k)}\right)\right), \quad t = 0, \dots, T-1, \tag{8}$$

where $\epsilon > 0$ is a numerical stabilizer and $\odot$ denotes element-wise multiplication.

After refinement, we obtain the final pseudo-label $\tilde{\mathbf{p}}_j^u = \tilde{\mathbf{p}}_j^{(T)}$. We then perform one fresh forward pass of $\mathbf{x}_j^u$ to obtain prediction $\mathbf{p}_j^u \in \Delta^C$, and define the pseudo-label loss using KL divergence:

$$\mathcal{L}_{\text{pseudo}} = \frac{1}{B_u} \sum_{j=1}^{B_u} \text{KL}\left(\tilde{\mathbf{p}}_j^u \| \mathbf{p}_j^u\right) = \frac{1}{B_u} \sum_{j=1}^{B_u} \sum_{c=1}^{C} \tilde{\mathbf{p}}_{j,c}^u \log \frac{\tilde{\mathbf{p}}_{j,c}^u}{\mathbf{p}_{j,c}^u}. \tag{9}$$

Finally, the total training loss aggregates the supervised and pseudo-label objectives:

$$\mathcal{L}_{\text{total}} = \mathcal{L}_{\text{sup}} + \lambda \cdot \mathcal{L}_{\text{pseudo}}, \tag{10}$$

where $\lambda$ is a balancing hyperparameter.

This formulation enables AKOrN to perform efficient, augmentation-free semi-supervised learning by distilling structure-aware pseudo-labels from low-energy oscillatory dynamics.**Energize** acts as a refinement layer on top of the Kuramoto backbone, guiding the model toward robust consensus predictions under uncertainty and noise.

## 4 THEORETICAL ANALYSIS

A central question in our framework is whether energy-aware fusion of stochastic predictions yields a more reliable pseudo-label distribution than individual noisy predictions. Intuitively, low-energy states of AKOrN correspond to stable synchronization and thus more trustworthy signals. By weighting predictions according to their energies, we aim to reduce divergence from the true but unknown label distribution. The following theorem formalizes this intuition.

**Theorem 4.1.** *Let $\mathbf{p}^*$ be the true (but unknown) class probability distribution of a sample. Through stochastic forward passes of the AKOrN, we obtain $K$ noisy estimates $\{\mathbf{p}^{(k)}\}_{k=1}^K$ of $\mathbf{p}^*$ and their corresponding energies $\{E^{(k)}\}_{k=1}^K$. Define the energy weights as:*

$$w^{(k)} = \frac{\exp(-\beta E^{(k)})}{\sum_{l=1}^{K} \exp(-\beta E^{(l)})}, \tag{11}$$

*where $\beta > 0$ is a temperature hyperparameter. Then, the fused pseudo-label*

$$\tilde{\mathbf{p}} = \sum_{k=1}^{K} w^{(k)} \mathbf{p}^{(k)}, \tag{12}$$

*satisfies*

$$\mathbb{E}\left[KL(\tilde{\mathbf{p}} \| \mathbf{p}^*)\right] \leq \sum_{k=1}^{K} w^{(k)} \cdot \mathbb{E}\left[KL(\mathbf{p}^{(k)} \| \mathbf{p}^*)\right]. \tag{13}$$

Theorem 4.1 shows that energy-weighted fusion yields a pseudo-label distribution closer in expectation to the true distribution than a simple average of stochastic components. This supports our design of **Energize**, where the AKOrN energy landscape not only guides pseudo-label selection but also improves calibration and robustness. Thus, energy-aware aggregation produces more reliable pseudo-labels, enhancing the stability and generalization of **Kuromi** without extra data augmentation or handcrafted thresholds. The proof is given in Appendix A.1.

Building on this result, we analyze the iterative refinement in **Energize**. While Theorem 4.1 shows that one fusion step improves pseudo-label reliability, **Energize** recursively aligns pseudo-labels with low-energy predictions. The next theorem establishes convergence of this update.

**Theorem 4.2.** *Let $\{\mathbf{p}^{(k)}\}_{k=1}^{K} \subset \Delta^C$ denote $K$ class-probability vectors with associated positive weights $\{w^{(k)}\}_{k=1}^{K}$, where $w^{(k)} = \frac{1}{E^{(k)}+\epsilon}$ and $\epsilon > 0$. Define the mapping $F : \Delta^C \to \Delta^C$ from Eq. 8 by*

$$F(\mathbf{q}) = Normalize\left(\sum_{k=1}^{K} w^{(k)} \cdot (\mathbf{q} \odot \mathbf{p}^{(k)})\right), \tag{14}$$

*where $\odot$ denotes element-wise multiplication and $\Delta^C = \{\mathbf{q} \in \mathbb{R}^C : q_i \geq 0, \sum_{i=1}^{C} q_i = 1\}$ is the probability simplex. Then the sequence $\{\tilde{\mathbf{p}}^{(t)}\}_{t=0}^{\infty}$ defined recursively by*

$$\tilde{\mathbf{p}}^{(t+1)} = F(\tilde{\mathbf{p}}^{(t)}), \tag{15}$$

*converges to a fixed point $\tilde{\mathbf{p}}^* \in \Delta^C$, i.e.,*

$$\lim_{t \to \infty} \tilde{\mathbf{p}}^{(t)} = \tilde{\mathbf{p}}^*, \quad where \ \tilde{\mathbf{p}}^* = F(\tilde{\mathbf{p}}^*), \tag{16}$$

*at which it achieves a stable agreement with all weighted predictions $\mathbf{p}^{(k)}$.*

The proof is given in Appendix A.2. Theorem 4.2 shows that the **Energize** refinement is stable and consistent: from any initialization, updates converge to an equilibrium distribution balancing all low-energy predictions. Together with Theorem 4.1, this establishes the theoretical basis for **Kuromi**'s augmentation-free SSL, where the energy landscape drives fusion and refinement, yielding increasingly reliable pseudo-labels without external heuristics.

| Method | CIFAR-10 (4k) | SVHN (1k) |
|---|---|---|
| FixMatch Sohn et al. (2020) | 94.3 | 92.1 |
| UDA Xie et al. (2020) | 93.4 | 91.2 |
| FlexMatch Zhang et al. (2021) | 94.9 | 92.7 |
| Meta Pseudo Label Pham et al. (2021) | 95.1 | 93.5 |
| Semi-ViM-Base He et al. (2025) | 95.8 | 94.3 |
| **Kuromi (Ours)** | **96.5** | **95.0** |

Table 1: Test accuracy (%) on CIFAR-10 and SVHN with limited labeled data (4k and 1k respectively). **Kuromi** achieves new state-of-the-art results.

| Method | STL-10 |
|---|---|
| FixMatch Sohn et al. (2020) | 89.5 |
| UDA Xie et al. (2020) | 88.6 |
| FlexMatch Zhang et al. (2021) | 90.1 |
| Meta Pseudo Label Pham et al. (2021) | 90.6 |
| Semi-ViM-Base He et al. (2025) | 91.3 |
| **Kuromi (Ours)** | **92.8** |

Table 2: Test accuracy (%) on STL-10 using full labeled data and 100k unlabeled samples. **Kuromi** achieves state-of-the-art performance.

## 5 EVALUATION RESULTS

**Datasets.** We evaluate **Kuromi** on four standard SSL benchmarks: CIFAR-10 (Krizhevsky (2009)), SVHN (Netzer et al. (2011)), STL-10 (Coates et al. (2011)), and ImageNet (Deng et al. (2009)). Full dataset descriptions are provided in Appendix F.

**Implementation Details. Kuromi** is implemented in PyTorch and trained on a single NVIDIA A100 GPU. Full hyperparameter settings and training configurations are provided in Appendix F.

Table 1 and Table 2 report **Kuromi**'s performance on CIFAR-10, SVHN, and STL-10 under low labeled data settings. **Kuromi** achieves 96.5% on CIFAR-10 (4k labels), 95.0% on SVHN (1k labels), and 92.8% on STL-10 (100k unlabeled), consistently surpassing prior methods. In Table 3, we compare **Kuromi** against state-of-the-art methods under different label ratios on ImageNet. Among lightweight and convolutional models, **Kuromi** significantly outperforms FixMatch (62.3% vs. 52.6% at 1% labels), and remains competitive with transformer-based approaches. With **Energize**, performance further improves to 65.1%, setting a new record among non-transformer models without data augmentation. With 10% and 25% of the labels, **Kuromi** (+Energize) continues to outperform methods like SimCLRv2+KD and Meta Pseudo Label. While ViT-based models such as REACT achieve higher absolute accuracy, they depend on large backbones and heavy pre-training. In contrast, **Kuromi** provides a lightweight, augmentation-free alternative that scales well with limited labels.

Compared to existing semi-supervised methods, **Kuromi** offers superior computational and inference efficiency. As shown in Table 4, it achieves **2200 images/sec** with just **12M parameters** and **0.18 GFLOPs**, surpassing all baselines by a wide margin. In contrast, methods like FixMatch and SimCLRv2 rely on heavy ResNet backbones, while transformer-based models such as Semi-ViM-Base require over **12× more computations** and run at less than **1/3 the speed**. These results underscore the hardware efficiency of the AKOrN backbone, making **Kuromi** ideal for low-resource or real-time SSL scenarios.

Table 3: Top-1 accuracy (%) on ImageNet under 1%, 10%, and 25% labeled data settings. All methods are evaluated using their official backbone configurations.

| Model | Backbone | Type | 1% | 10% | 25% |
|---|---|---|---|---|---|
| FixMatch Sohn et al. (2020) | ResNet-50 | Consistency + Pseudo-labeling | 52.6 | 68.7 | 74.9 |
| UDA Xie et al. (2020) | ResNet-50 | Consistency + DistAlign | 51.2 | 67.5 | 73.8 |
| FlexMatch Zhang et al. (2021) | WideResNet-28 | Confidence-aware Threshold | 53.5 | 70.2 | 75.3 |
| Meta Pseudo Label Pham et al. (2021) | EfficientNet-L2 | Meta-learning | 55.0 | 71.8 | 76.4 |
| SimCLRv2+KD Chen et al. (2020) | ResNet-50 | Self-supervised + KD | 54.5 | 69.7 | 75.5 |
| Semi-ViT (B) Cai et al. (2022) | ViT-B/16 | Self-training | 74.1 | 81.6 | 84.2 |
| Semi-ViT (L) Cai et al. (2022) | ViT-L/16 | Self-training | 77.3 | 83.3 | 85.1 |
| Semi-ViT (H) Cai et al. (2022) | ViT-H/14 | Self-training | 78.9 | 84.6 | 86.2 |
| REACT Liu et al. (2023) | ViT-L/16 | Robust SSL | 81.6 | 85.1 | 86.8 |
| SemiFormer Weng et al. (2022) | ViT-B/16 | Semi-supervised ViT | 75.8 | 82.1 | 84.5 |
| DINO Fini et al. (2023) | ViT-L/16 | Self-supervised | 78.1 | 82.9 | 84.9 |
| Co-Training Rothenberger & Diochnos (2023) | MLP-Mixer | Co-training | 80.1 | 85.1 | - |
| Semi-ViM-Base He et al. (2025) | ViM-Base | LyapEMA | **83.5** | **87.4** | **88.6** |
| Kuromi (w/o Energize) | AKOrN | Energy-based + No Aug | 62.3 | 74.2 | 78.4 |
| **Kuromi (+Energize)** | AKOrN | + Label Refinement | **65.1** | **76.0** | **80.1** |

Table 4: Model size, compute, and throughput comparison on ImageNet under a 1% labeled data setting. **Kuromi** achieves the highest efficiency in terms of both FLOPs and processing speed.

| Model | #Params (M) | FLOPs (G) | Throughput (img/s) |
|---|---|---|---|
| FixMatch (ResNet-50) Sohn et al. (2020) | 25.0 | 1.8 | 1250 |
| SimCLRv2-distill (ResNet-50) Chen et al. (2020) | 28.0 | 1.8 | 1180 |
| SimCLRv2-self (ResNet-152) Chen et al. (2020) | 28.0 | 4.1 | 960 |
| Co-Training (MLP) Rothenberger & Diochnos (2023) | 56.0 | 2.6 | 980 |
| Semi-ViT-Base Cai et al. (2022) | 146.0 | 3.2 | 670 |
| **Kuromi (Ours)** | **12.0** | **0.18** | **2200** |

## 5.1 FEW-SHOT RESULTS

To further examine **Kuromi**'s representation quality in low-data regimes, we conduct a qualitative analysis using the MiniImageNet dataset under a few-shot setting. Specifically, we randomly sample 10 images per class from 5 test categories, and visualize the learned features using t-SNE. As shown in Fig. 2, **Kuromi** produces well-separated and compact clusters, indicating strong semantic discrimination even under limited supervision. In contrast, FixMatch exhibits overlapping clusters, and Semi-ViT-Base struggles to maintain a margin between neighboring classes. These results highlight **Kuromi**'s ability to learn structured latent spaces from sparse supervision, benefiting few-shot generalization. The corresponding quantitative results are reported in Table 17, confirming **Kuromi**'s superior performance in few-shot generalization.

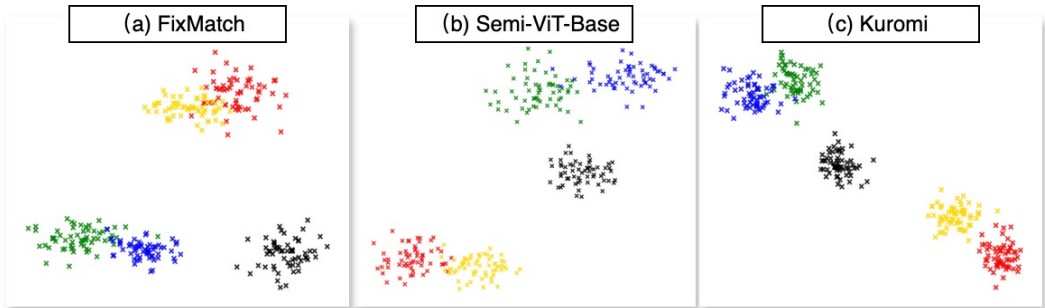

Figure 2: t-SNE visualization on MiniImageNet (5-way, 10-shot). **Kuromi** learns more compact and well-separated class clusters compared to FixMatch and Semi-ViT-Base.

## 5.2 ABLATION STUDIES

As shown in Fig. 3, **Kuromi** exhibits strong robustness and clear modular contributions. The accuracy in Fig. 3(a) increases with the number of stochastic forward passes $K$ and saturates around $K=5$, validating the benefit of multi-view consensus in **Energize**. Fig. 3(b) further disentangles the role of each component. Removing the oscillatory dynamics ("w/o oscillator") replaces each oscillator state $x_i = (\cos\theta_i, \sin\theta_i)$ with a standard real-valued feature, eliminating phase synchronization and energy descent; this substantially weakens stability and leads to a 4–5% accuracy drop. Similarly, discarding energy-based selection ("w/o energy selection") retains the $K$ stochastic passes but fuses them through uniform averaging rather than weighting by $(E^{(k)} + \varepsilon)^{-1}$, thereby ignoring dynamical stability information and producing a comparable degradation. Using random pseudo-labels incurs the largest drop (7.5%), highlighting the necessity of energy-guided refinement. Finally, Fig. 3(c) shows that **Kuromi** scales effectively with varying label budgets, maintaining strong accuracy even with only 500 labels and achieving over 96% once 4k labels are available.

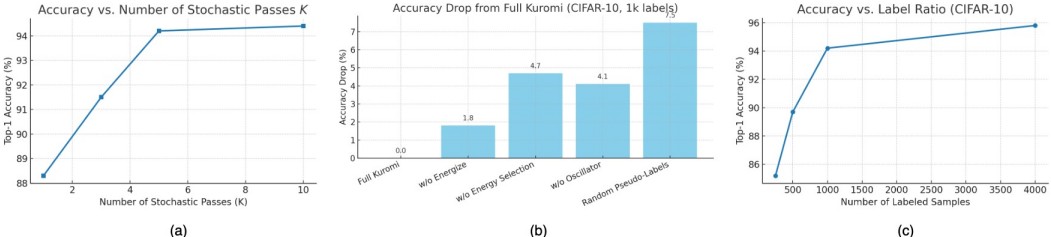

(a)          (b)          (c)

Figure 3: **Ablation Analysis of Kuromi.** (a) Accuracy vs. stochastic passes $K$: performance improves with multi-view sampling and saturates at $K = 5$. (b) Component ablation on CIFAR-10 (1k labels): removing **Energize**, energy selection, or oscillatory dynamics reduces accuracy; random pseudo-labeling performs worst. (c) Accuracy vs. label ratio: **Kuromi** scales well, maintaining strong accuracy even with 500 labels.

Table 5: Comparison of pseudo-label quality metrics on CIFAR-10 (1k labeled). Lower entropy and ECE indicate more confident and better-calibrated predictions. Higher agreement score reflects more consistent pseudo-labels across stochastic passes.

| Method | Entropy ↓ | ECE (%) ↓ | Agreement (%) ↑ |
|---|---|---|---|
| Random Pseudo-Labels | 2.33 | 12.8 | 37.2 |
| Confidence Threshold (FixMatch) | 1.21 | 6.5 | 61.0 |
| Energy-based (1-shot) | 0.96 | 4.3 | 72.5 |
| **Kuromi + Energize** | **0.72** | **2.6** | **85.4** |

Table 6: Comparison of pseudo-label refinement strategies on CIFAR-10 (1k labels). **Energize** yields higher accuracy than simple averaging.

Table 7: Effect of the number of **Energize** refinement steps $T_{\text{refine}}$ on CIFAR-10.

| Strategy | Refine Type | Acc. (%) |
|---|---|---|
| Simple Avg | Uniform, no iteration | 95.4 |
| **Energize** | Energy-weighted, iterative | **96.5** |

| $T_{\text{refine}}$ | Acc. (%) | Time (ms) |
|---|---|---|
| 1 | 95.8 | 1.0 |
| 2 | 96.2 | 1.8 |
| 3 | **96.5** | 2.5 |
| 5 | 96.4 | 4.1 |

Table 5 shows that pseudo-labels refined by **Energize** have significantly lower entropy and calibration error (ECE), as well as higher agreement across stochastic predictions. This confirms that **Kuromi** yields more reliable supervisory signals, which in turn improves semi-supervised performance.

To evaluate the effectiveness of our proposed pseudo-label refinement mechanism, we compare **Energize** with a simple averaging baseline. As shown in Table 6, **Energize** achieves a notable improvement of 1.1% in accuracy over uniform averaging by incorporating both energy-based weighting and iterative denoising. Unlike uniform averaging, which assigns equal weight to all $K$ stochastic predictions and therefore allows unstable high-energy trajectories to influence the fused pseudo-labels, **Energize** automatically downweights such unreliable predictions through $(E^{(k)} + \varepsilon)^{-1}$

scaling and refines them through consistency-driven smoothing. We further study the impact of the number of refinement steps $T_{\text{refine}}$, which controls how many times the prediction smoothing is iteratively applied, on performance and computational cost. As shown in Table 7, accuracy steadily improves with more refinement iterations, peaking at three steps. Importantly, even a single-step refinement already surpasses the non-refined baseline, while adding excessive iterations brings diminishing returns and increased runtime. These results validate the efficiency and robustness of **Energize**, demonstrating its effectiveness without introducing sensitive hyperparameters.

## 6 CONCLUSION

In this paper, we proposed **Kuromi**, a novel framework that offers a new perspective on SSL by replacing data augmentation and teacher-driven heuristics with dynamic neuron interactions and energy-based refinement. Built on the lightweight AKOrN architecture and the novel **Energize** mechanism, it achieves high accuracy with minimal compute. Looking ahead, we aim to reduce the cost of stochastic inference and extend **Kuromi** to broader domains including pre-training, multimodal learning, and efficient on-device deployment.

## ETHICS STATEMENT

Our research proposes a semi-supervised learning framework, **Kuromi**, that eliminates the need for data augmentation and enhances learning from scarce annotations. The datasets used in this study—CIFAR-10/100, STL-10, SVHN, MiniImageNet, and ImageNet-1K—are all publicly available and widely used in the community. No personally identifiable information or human subjects are involved in this work. We do not release any new dataset as part of this study.

Our method does not involve potentially harmful applications, such as biometric identification or surveillance, nor is it intended for use in safety-critical domains without further domain-specific calibration. The proposed approach may benefit scenarios where data annotation is expensive or sensitive, including low-resource or ecologically important settings. However, we caution that the use of learned models should always account for domain bias and fairness considerations inherent to the training data.

All co-authors confirm that the research complies with the ICLR Code of Ethics. There are no conflicts of interest, undisclosed sponsorships, or ethical concerns regarding our methodology or findings.

## REPRODUCIBILITY STATEMENT

To ensure the reproducibility of our results, we provide a comprehensive implementation of **Kuromi** in the supplementary material, including all training code, model architectures, hyperparameter settings, and evaluation scripts. Key reproducibility components include:

- Full algorithmic details of the **AKOrN** backbone and the **Energize** module are provided in main text and Appendix C.
- Hyperparameters and training configurations (e.g., optimizer, learning rate schedules, batch sizes) for all datasets are listed in Appendix F.
- Datasets and preprocessing pipelines are publicly available, and we detail our usage in Appendix F and G.
- Our codebase includes ablation toggles and deterministic seeding for all experiments.

We believe these efforts ensure that other researchers can verify and extend our work with minimal overhead.

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

# A  THEORETICAL ANALYSIS

In this section, we provide the proof of Theorem 4.1 and Theorem 4.2.

## A.1  PROOF OF THEOREM 4.1

*Proof.* For probability distributions $p = \{p(i)\}_{i \in \mathcal{I}}$ and $q = \{q(i)\}_{i \in \mathcal{I}}$ (with $p(i) > 0$ whenever $q(i) > 0$) on a finite or countable set $\mathcal{I}$, the KL divergence is defined as

$$\text{KL}(p \parallel q) = \sum_{i \in \mathcal{I}} p(i) \log p(i) - \sum_{i \in \mathcal{I}} p(i) \log q(i). \tag{17}$$

First, we want to calculate the first term of Equation 17. Let $\{p_k\}_{k=1}^{K}$ be the probability distributions on $\mathcal{I}$ and let $\{w^{(k)}\}_{k=1}^{K}$ be the weights satisfying $w^{(k)} \geq 0$ and $\sum_{k=1}^{K} w^{(k)} = 1$. For each $i \in \mathcal{I}$ we have

$$\sum_{k=1}^{K} w^{(k)} p_k(i) \log\Big(p_k(i)\Big) \geq \left(\sum_{k=1}^{K} w^{(k)} p_k(i)\right) \log\left(\frac{\sum_{k=1}^{K} w^{(k)} p_k(i)}{1}\right), \tag{18}$$

by Jensen's inequality Jensen (1906) applied to the convex function $f(x) = x \log x$. Rearranging, we obtain

$$\left(\sum_{k=1}^{K} w^{(k)} p_k(i)\right) \log\left(\sum_{k=1}^{K} w^{(k)} p_k(i)\right) \leq \sum_{k=1}^{K} w^{(k)} p_k(i) \log p_k(i). \tag{19}$$

Define the mixture distribution $p$ by

$$p(i) = \sum_{k=1}^{K} w^{(k)} p_k(i), \quad \text{for each } i \in \mathcal{I}. \tag{20}$$

Then, by summing the inequality over all $i \in \mathcal{I}$, we have

$$\begin{aligned}
\sum_{i \in \mathcal{I}} p(i) \log p(i) &\leq \sum_{i \in \mathcal{I}} \sum_{k=1}^{K} w^{(k)} p_k(i) \log p_k(i) \\
&= \sum_{k=1}^{K} w^{(k)} \sum_{i \in \mathcal{I}} p_k(i) \log p_k(i).
\end{aligned} \tag{21}$$

Next, we want to calculate the second term of Equation 17. Since it is linear in $p$, we have

$$\begin{aligned}
\sum_{i \in \mathcal{I}} p(i) \log q(i) &= \sum_{i \in \mathcal{I}} \left(\sum_{k=1}^{K} w^{(k)} p_k(i)\right) \log q(i) \\
&= \sum_{k=1}^{K} w^{(k)} \sum_{i \in \mathcal{I}} p_k(i) \log q(i).
\end{aligned} \tag{22}$$

Similarly, for each $k$:

$$\text{KL}(p_k \parallel q) = \sum_{i \in \mathcal{I}} p_k(i) \log p_k(i) - \sum_{i \in \mathcal{I}} p_k(i) \log q(i). \tag{23}$$

Multiplying by $w^{(k)}$ and summing over $k$:

$$\begin{aligned}
\sum_{k=1}^{K} w^{(k)} \text{KL}(p_k \parallel q) &= \sum_{k=1}^{K} w^{(k)} \left[\sum_{i \in \mathcal{I}} p_k(i) \log p_k(i) - \sum_{i \in \mathcal{I}} p_k(i) \log q(i)\right]. \\
&= \sum_{k=1}^{K} w^{(k)} \sum_{i \in \mathcal{I}} p_k(i) \log p_k(i) - \sum_{k=1}^{K} w^{(k)} \sum_{i \in \mathcal{I}} p_k(i) \log q(i).
\end{aligned} \tag{24}$$

By inequality 21, 22, and Eq. 24, the linear terms involving $\log q(i)$ are equal on both sides. Hence,

$$\mathrm{KL}\Big(\sum_{k=1}^{K} w^{(k)} p_k \parallel q\Big) \leq \sum_{k=1}^{K} w^{(k)} \mathrm{KL}(p_k \parallel q). \tag{25}$$

By identifying $\tilde{\mathbf{p}} = p$, $\quad \mathbf{p}^{(k)} = p_k$, $\quad \mathbf{p}^* = q$, we conclude that for any set of predictions $\{\mathbf{p}^{(k)}\}_{k=1}^{K}$ and corresponding weights $w^{(k)}$:

$$\mathrm{KL}(\tilde{\mathbf{p}} \parallel \mathbf{p}^*) \leq \sum_{k=1}^{K} w^{(k)} \mathrm{KL}(\mathbf{p}^{(k)} \parallel \mathbf{p}^*). \tag{26}$$

By inequality 26, it holds for every set of stochastic initializations (i.e., for every realization of $\{\mathbf{p}^{(k)}\}$), taking the expectation over these realizations gives:

$$\mathbb{E}\left[\mathrm{KL}(\tilde{\mathbf{p}} \parallel \mathbf{p}^*)\right] \leq \mathbb{E}\left[\sum_{k=1}^{K} w^{(k)} \mathrm{KL}(\mathbf{p}^{(k)} \parallel \mathbf{p}^*)\right]. \tag{27}$$

Let $\{w^{(k)}\}_{k=1}^{K}$ and $\{\mathbf{p}^{(k)}\}_{k=1}^{K}$ be collections of random variables, where each weight $w^{(k)}$ depends (possibly in a complicated way) on the corresponding energy $E^{(k)}$ and the outputs $\mathbf{p}^{(k)}$. Define the quantity

$$X^{(k)} := w^{(k)} \mathrm{KL}\Big(\mathbf{p}^{(k)} \parallel \mathbf{p}^*\Big) \tag{28}$$

as a random variable for each $k = 1, 2, \ldots, K$.

By the linearity of the expectation operator, substituting back $X^{(k)} = w^{(k)} \mathrm{KL}(\mathbf{p}^{(k)} \parallel \mathbf{p}^*)$, we obtain:

$$\mathbb{E}\left[\sum_{k=1}^{K} w^{(k)} \mathrm{KL}(\mathbf{p}^{(k)} \parallel \mathbf{p}^*)\right] = \sum_{k=1}^{K} \mathbb{E}\left[w^{(k)} \mathrm{KL}(\mathbf{p}^{(k)} \parallel \mathbf{p}^*)\right]. \tag{29}$$

then, we obtain:

$$\begin{aligned} \mathbb{E}\left[\mathrm{KL}\Big(\tilde{\mathbf{p}} \parallel \mathbf{p}^*\Big)\right] &\leq \mathbb{E}\left[\sum_{k=1}^{K} w^{(k)} \mathrm{KL}\Big(\mathbf{p}^{(k)} \parallel \mathbf{p}^*\Big)\right] \\ &= \sum_{k=1}^{K} \mathbb{E}\left[w^{(k)} \mathrm{KL}\Big(\mathbf{p}^{(k)} \parallel \mathbf{p}^*\Big)\right]. \end{aligned} \tag{30}$$

$\square$

### A.2 PROOF OF THEOREM 4.2

*Proof.* We have the mapping $F : \Delta^C \to \Delta^C$, with the probability simplex

$$\Delta^C = \Big\{\mathbf{q} \in \mathbb{R}^C \Big| q_i \geq 0, \sum_{i=1}^{C} q_i = 1\Big\}, \tag{31}$$

by

$$F(\mathbf{q}) = \mathrm{Normalize}\left(\sum_{k=1}^{K} \alpha^{(k)} (\mathbf{q} \odot \mathbf{p}^{(k)})\right), \quad \text{where} \quad \alpha^{(k)} = \frac{1}{E^{(k)} + \epsilon}. \tag{32}$$

In other words, if we define $f(\mathbf{q}) = \sum_{k=1}^{K} \alpha^{(k)} (\mathbf{q} \odot \mathbf{p}^{(k)})$, then $F(\mathbf{q}) = \frac{f(\mathbf{q})}{\|f(\mathbf{q})\|_1}$, so that the iteration is $\mathbf{q}^{(t+1)} = F(\mathbf{q}^{(t)})$.

Let $d(\mathbf{q}_1, \mathbf{q}_2)$ denote a metric on $\Delta^C$; for example, we may choose the Euclidean norm:

$$d(\mathbf{q}_1, \mathbf{q}_2) = \|\mathbf{q}_1 - \mathbf{q}_2\|_2. \tag{33}$$

Our goal is to show that there exists a constant $L \in [0, 1)$ such that for all $\mathbf{q}_1, \mathbf{q}_2 \in \Delta^C$,

$$d\big(F(\mathbf{q}_1), F(\mathbf{q}_2)\big) \leq Ld\big(\mathbf{q}_1, \mathbf{q}_2\big). \tag{34}$$

Once such an $L$ is found, the Banach Fixed-Point Theorem Banach (1932) guarantees that a unique fixed point $\mathbf{q}^*$ exists and that the sequence $\{\mathbf{q}^{(t)}\}$ converges to $\mathbf{q}^*$.

Notice that for any two points $\mathbf{q}_1, \mathbf{q}_2 \in \Delta^C$ and for each coordinate $i = 1, \ldots, C$,

$$f_i(\mathbf{q}) = \sum_{k=1}^{K} \alpha^{(k)} q_i p_i^{(k)}. \tag{35}$$

Because for fixed $k$ and $i$, $p_i^{(k)}$ and $\alpha^{(k)}$ are constants, the mapping $q_i \mapsto \alpha^{(k)} q_i p_i^{(k)}$ is linear. Thus for each coordinate we have:

$$
\begin{aligned}
|f_i(\mathbf{q}_1) - f_i(\mathbf{q}_2)| &= \left| \sum_{k=1}^{K} \alpha^{(k)} p_i^{(k)} \Big( q_{1,i} - q_{2,i} \Big) \right| \\
&\leq \Big( \sum_{k=1}^{K} \alpha^{(k)} p_i^{(k)} \Big) |q_{1,i} - q_{2,i}|.
\end{aligned}
\tag{36}
$$

Taking the vector 2-norm and using standard inequalities, there exists a constant $L_f > 0$ such that

$$\|f(\mathbf{q}_1) - f(\mathbf{q}_2)\|_2 \leq L_f \|\mathbf{q}_1 - \mathbf{q}_2\|_2. \tag{37}$$

The constant $L_f$ may be bounded in terms of the parameters $\alpha^{(k)}$ and the entries of $\mathbf{p}^{(k)}$. Since all $\mathbf{p}^{(k)}$ are probability vectors, the entries satisfy $0 \leq p_i^{(k)} \leq 1$.

Assume there exists a uniform lower bound $m > 0$ such that for all $\mathbf{q} \in \Delta^C$ and for every coordinate, $f_i(\mathbf{q}) \geq m$, so that $\|f(\mathbf{q})\|_1 \geq Cm$. This condition is reasonable if the entries of $\mathbf{p}^{(k)}$ are strictly positive and the weights $\alpha^{(k)}$ are bounded. Under this condition, we can prove via the mean value theorem for vector-valued functions Rudin (1976) that there exists a constant $L_N < \infty$ such that

$$\|F(\mathbf{q}_1) - F(\mathbf{q}_2)\|_2 \leq L_N \|f(\mathbf{q}_1) - f(\mathbf{q}_2)\|_2. \tag{38}$$

Combine inequality 37 and inequality 38, we obtain

$$\|F(\mathbf{q}_1) - F(\mathbf{q}_2)\|_2 \leq (L_N L_f) \|\mathbf{q}_1 - \mathbf{q}_2\|_2. \tag{39}$$

Define $L = L_N L_f$. If we can show that $L < 1$, then for every $\mathbf{q}_1, \mathbf{q}_2 \in \Delta^C$,

$$\|F(\mathbf{q}_1) - F(\mathbf{q}_2)\|_2 \leq L \|\mathbf{q}_1 - \mathbf{q}_2\|_2, \tag{40}$$

i.e., $F$ is a contraction mapping. Then the Banach Fixed-Point Theorem guarantees that $F$ has a unique fixed point $\mathbf{q}^* \in \Delta^C$ and that for any initialization $\mathbf{q}^{(0)} \in \Delta^C$ the iteration

$$\mathbf{q}^{(t+1)} = F(\mathbf{q}^{(t)}) \tag{41}$$

converges to $\mathbf{q}^*$.

The condition $L < 1$ depends on the properties of the weights $\alpha^{(k)}$ and the agreement multipliers $p_i^{(k)}$. In practice, the experiments (see Table 7 in the paper) indicate that a few iterations (typically $T = 3$) suffice to reach stability, which provides strong numerical support that $L$ is indeed less than 1, even if an exact analytical bound may be hard to derive in full generality.

$\square$

## B  RELATION TO ENERGY-BASED MODELS

Energy-based models (EBMs) have a long history in machine learning, dating back to the Hopfield network and Boltzmann machines. Early studies introduced the foundational perspective that learning can be framed as minimizing a scalar *energy function* that measures compatibility between inputs and outputs (LeCun et al., 2006). Later works proposed *Equilibrium Propagation* (Scellier & Bengio,

2017), showing that energy-based dynamics can compute gradients through local perturbations, bridging EBMs and biologically plausible learning. More recent efforts extended this idea to *latent variable EBMs*, advocating hierarchical predictive architectures and differentiable world models as building blocks toward autonomous intelligence (LeCun, 2022; Dawid & LeCun, 2024).

In parallel, subsequent advances demonstrated that EBMs can perform implicit generation through gradient-based dynamics, connecting the energy landscape to diffusion and score-matching methods (Du & Mordatch, 2019; Song & Ermon, 2019). Further analyses revealed that modern discriminative classifiers implicitly define energy-based distributions (*JEM*) (Grathwohl et al., 2020), highlighting the ubiquity of energy structures in deep networks.

Our approach, **Kuromi**, is conceptually related to these lines of research in that it leverages a structured energy landscape for inference. However, it departs from classical EBMs in two fundamental aspects. First, instead of defining a static scalar energy $E_\theta(x, y)$ and minimizing it via gradient descent, Kuromi employs a *dynamic oscillator system* that evolves neuron states $\mathbf{x}_i$ on a hypersphere:

$$\Delta\mathbf{x}_i = \Omega_i\mathbf{x}_i + \text{Proj}_{\mathbf{x}_i}\left(c_i + \sum_j J_{ij}\mathbf{x}_j\right), \quad \mathbf{x}_i \leftarrow \text{Normalize}(\mathbf{x}_i + \gamma\Delta\mathbf{x}_i),$$

where $\Omega_i$ denotes intrinsic rotation and $J_{ij}$ denotes coupling strength. This formulation implicitly minimizes an energy functional

$$\mathcal{E} = -\sum_{i,j} J_{ij}\,\mathbf{x}_i^\top\mathbf{x}_j,$$

but the minimization arises through *temporal synchronization*, not explicit backpropagation through $E_\theta(x, y)$.

Second, Kuromi performs *amortized energy inference* via the *Energize* module, which aggregates multiple stochastic trajectories into a low-energy consensus state. This contrasts with Langevin or contrastive divergence methods in EBMs, where energy minimization is achieved through stochastic sampling or MCMC.

In summary, Kuromi bridges energy-based reasoning and oscillatory computation: it inherits the inference-by-descent intuition from EBMs but replaces explicit energy minimization with emergent synchronization dynamics, enabling lightweight and augmentation-free semi-supervised learning.

## C  PSEUDOCODE OF ENERGIZE

---
**Algorithm 1** Energize: Stability-Aware Energy-Weighted Pseudo-Label Refinement
---
1: **Input:** unlabeled sample $x$, model $f_\theta$ (AKOrN + classifier), number of stochastic views $K$, refinement steps $T_{\text{refine}}$, stabilizer $\epsilon$
2: **Output:** refined pseudo-label $\tilde{p}$, pseudo loss $L_{\text{pseudo}}$
3: **for** $k = 1$ to $K$ **do**
4:     Initialize oscillator states and run stochastic forward pass: $(p^{(k)}, E^{(k)}) \leftarrow$ STOCHASTICFORWARD$(f_\theta, x)$
5: **end for**
6: Initialize pseudo-label: $\tilde{p} \leftarrow$ NORMALIZE$\left(\sum_{k=1}^{K} \frac{1}{E^{(k)} + \epsilon} p^{(k)}\right)$
7: **for** $t = 1$ to $T_{\text{refine}}$ **do**
8:     $q \leftarrow \mathbf{0}$
9:     **for** $k = 1$ to $K$ **do**
10:         $q \leftarrow q + \frac{1}{E^{(k)} + \epsilon}(\tilde{p} \odot p^{(k)})$
11:     **end for**
12:     $\tilde{p} \leftarrow$ NORMALIZE$(q)$
13: **end for**
14: $(p_u, \_) \leftarrow$ STOCHASTICFORWARD$(f_\theta, x)$
15: $L_{\text{pseudo}} \leftarrow \text{KL}(\tilde{p} \,\|\, p_u) = \sum_c \tilde{p}_c \log \frac{\tilde{p}_c}{p_{u,c}}$
16: **return** $\tilde{p}, L_{\text{pseudo}}$
---

The ENERGIZE algorithm refines pseudo-labels for unlabeled samples using energy-aware stochastic predictions. For a given sample $x$, $K$ stochastic forward passes are performed through the model $f_\theta$, where randomness comes from initializing the oscillator states in the AKOrN backbone. Each pass produces a probability vector $p^{(k)}$ and a Kuramoto energy $E^{(k)}$, where lower energy indicates a more stable and coherent prediction. The initial pseudo-label $\tilde{p}$ is computed as an energy-weighted average of these predictions, with normalization ensuring it lies in the probability simplex $\Delta_C$. The label is then iteratively refined by combining element-wise products of $\tilde{p}$ with each stochastic prediction, again weighted by the inverse energy to emphasize agreement among low-energy predictions, following principles similar to consistency regularization in SSL. Finally, a fresh stochastic forward pass yields $p_u$, and the KL divergence between $\tilde{p}$ and $p_u$ serves as the pseudo-label loss $L_{\text{pseudo}}$, supervising the unlabeled sample while encouraging energy-consistent predictions.

## D    COMPUTATION COST ANALYSIS

To provide a clearer view of **Kuromi**'s efficiency, we report the detailed computational breakdown of each training stage and the overhead introduced by ENERGIZE. All experiments are conducted on a single NVIDIA A100 GPU.

### D.1    STAGE-WISE TRAINING TIME BREAKDOWN

Table 8 shows the percentage of wall-clock training time spent in each training stage. Pre-training accounts for the largest fraction due to contrastive loss computation, whereas the cost of ENERGIZE refinement is relatively small, confirming that **Kuromi** remains efficient even with iterative pseudo-label refinement.

Table 8: Stage-wise training time breakdown for CIFAR-10 (4k labels, 300 epochs).

| Stage | Time per Epoch (ms) | Share of Total Time (%) |
|---|---|---|
| AKOrN Pretraining | 145 | 52.3 |
| Supervised Fine-tuning | 72 | 25.9 |
| Pseudo-label Generation (K=5) | 41 | 14.9 |
| Energize Refinement ($T_{\text{refine}} = 3$) | 19 | 6.9 |

### D.2    IMPACT OF $K$ AND $T_{\text{REFINE}}$ ON ACCURACY AND RUNTIME

Table 9: Effect of $K$ (stochastic passes) and $T_{\text{refine}}$ (refinement steps) on CIFAR-10 (1k labels). Accuracy and runtime are averaged over 3 random seeds.

| $K$ | $T_{\text{refine}}$ | Top-1 Acc. (%) | Runtime per Epoch (ms) |
|---|---|---|---|
| 1 | 1 | 95.2 | 215 |
| 3 | 1 | 95.6 | 318 |
| 5 | 1 | 95.8 | 405 |
| 5 | 2 | 96.2 | 478 |
| 5 | 3 | **96.5** | 512 |
| 5 | 5 | 96.4 | 731 |

Table 9 extends Table 8 in the main paper by adding the total training time per epoch for each configuration. Increasing $K$ improves accuracy but introduces a near-linear increase in runtime. Similarly, larger $T_{\text{refine}}$ provides diminishing returns after three iterations, while nearly doubling the runtime when increased from $3 \rightarrow 5$. Overall, **Kuromi** achieves its best trade-off at $K = 5$ and $T_{\text{refine}} = 3$, which we adopt as the default setting throughout all experiments.

# E    MULTI-SEED ROBUSTNESS EVALUATION

Table 10: Mean and standard deviation of test accuracy (%) across five random seeds.

| Dataset | Kuromi (mean ± std) | Kuromi + ENERGIZE (mean ± std) |
|---|---|---|
| CIFAR-10 (4k labels) | $96.3 \pm 0.14$ | $\mathbf{96.5 \pm 0.12}$ |
| SVHN (1k labels) | $94.7 \pm 0.18$ | $\mathbf{95.0 \pm 0.15}$ |

To assess the stability of **Kuromi**, we repeat all CIFAR-10 and SVHN experiments using five independent random seeds for data splits, network initialization, and optimizer state. We report the mean and standard deviation of test accuracy across runs in Table 10. Results show consistently low variance ($< 0.2\%$), demonstrating that **Kuromi**'s performance is robust to stochasticity in training and data sampling. These results confirm that both the base **Kuromi** model and its ENERGIZE-enhanced variant are highly reproducible, with negligible run-to-run variability.

# F    TRAINING AND HYPERPARAMETER DETAILS

Table 11: Comprehensive training and hyperparameter settings.

| Component | Setting / Search Range (Final Choice) |
|---|---|
| Optimizer | Adam ($\beta_1 = 0.9, \beta_2 = 0.999, \epsilon = 10^{-8}$) |
| Learning Rate | $3 \times 10^{-4}$ with cosine decay scheduler |
| Warmup | Linear warmup for first 5 epochs |
| Weight Decay | $5 \times 10^{-5}$ |
| Batch Size | 512 (pretraining + semi-supervised training) |
| Kuramoto Steps $T$ | $\{4, 6, 8, 10\}$ (choose $T = 8$) |
| Stochastic Passes $K$ | $\{1, 3, 5, 7\}$ (choose $K = 5$) |
| Refinement Steps $T_{\text{refine}}$ | $\{1, 2, 3, 5\}$ (choose $T_{\text{refine}} = 3$) |
| Loss Weight $\lambda$ | Tuned on validation set: $\{0.5, 1.0, 2.0\}$ (choose $\lambda = 1.0$) |
| Epochs | CIFAR-10/SVHN/STL-10: 300; ImageNet: 100 |
| Early Stopping | Not used; best checkpoint selected at final epoch |

Table 13 summarizes all training configurations and hyperparameters used in our experiments, including optimizer settings, scheduler, and search ranges for key parameters. These settings are fixed across all datasets unless otherwise noted.

All experiments are run on a single NVIDIA A100 GPU. Unless otherwise stated, these hyperparameters remain consistent across datasets to ensure comparability.

Table 12: Summary of datasets and semi-supervised settings used in our experiments.

| Dataset | Image Size | #Classes | Labeled / Unlabeled | Label Ratio | Augmentation |
|---|---|---|---|---|---|
| CIFAR-10 | $32 \times 32$ | 10 | 1,000 / 49,000 | 2% | None |
| SVHN | $32 \times 32$ | 10 | 1,000 / 72,257 | 1.4% | None |
| STL-10 | $96 \times 96$ | 10 | 5,000 / 100,000 | 4.8% | None |
| ImageNet (1K) | $224 \times 224$ | 1,000 | 128,116 / 1,281,167 | 10% | None |

We summarize the hyperparameter settings used for training **Kuromi** across different benchmarks in Table 13. The number of Kuramoto evolution steps ($T$) and refinement iterations ($T_{\text{refine}}$) are kept modest (4–6 and 3, respectively) to balance dynamic modeling and efficiency. For stochastic

Table 13: Hyperparameter settings for Kuromi across different datasets.

| Parameter | CIFAR-10 | STL-10 | ImageNet-1K | MiniImageNet |
|---|---|---|---|---|
| Batch Size | 256 | 256 | 512 | 64 |
| Learning Rate ($\eta$) | 0.03 | 0.03 | 0.01 | 0.01 |
| Kuramoto Steps ($T$) | 4 | 4 | 6 | 6 |
| Refinement Steps ($T_{\text{refine}}$) | 3 | 3 | 3 | 3 |
| Stochastic Passes ($K$) | 5 | 5 | 7 | 5 |
| Weight Decay | 5e-4 | 5e-4 | 1e-4 | 1e-4 |
| Loss Weight ($\lambda$) | 1.0 | 1.0 | 0.5 | 1.0 |

pseudo-labeling, we use $K = 5$ forward passes on smaller datasets (e.g., CIFAR-10 and STL-10), and increase to $K = 7$ on ImageNet-1K to improve pseudo-label stability. The loss weight $\lambda$ controls the contribution of unlabeled data and is adjusted conservatively across datasets to avoid overfitting to pseudo-label noise. We use larger batch sizes and lower learning rates on ImageNet-1K to ensure convergence stability. Overall, our training pipeline uses consistent, reproducible hyperparameters without extensive tuning, and demonstrates strong performance across both low- and high-resolution settings.

Table 14: Optimizer and learning rate scheduler settings across datasets.

| Setting | CIFAR-10 | STL-10 | ImageNet-1K | MiniImageNet |
|---|---|---|---|---|
| Optimizer | SGD (momentum) | SGD (momentum) | AdamW | AdamW |
| Momentum / $\beta_1$ | 0.9 | 0.9 | 0.9 | 0.9 |
| Weight Decay | $5 \times 10^{-4}$ | $5 \times 10^{-4}$ | $1 \times 10^{-4}$ | $1 \times 10^{-4}$ |
| LR Scheduler | Cosine Annealing | Cosine Annealing | Linear Warmup + Cosine | Linear Warmup + Cosine |
| Warmup Epochs | – | – | 5 | 3 |
| Max Epochs | 300 | 300 | 200 | 100 |

We use momentum SGD for CIFAR-10 and STL-10 due to their smaller scale, while AdamW is adopted for ImageNet-1K and MiniImageNet to improve stability in high-resolution and low-data regimes. All settings employ cosine annealing schedules, optionally preceded by linear warm-up phases on larger datasets. These schedules ensure smooth convergence while reducing sensitivity to initial learning rate choices. See Table 14 for full optimizer and scheduler configurations.

**Dataset Details.** We evaluate **Kuromi** on four widely-used benchmarks for semi-supervised image classification: CIFAR-10, SVHN, STL-10, and ImageNet. Table 12 summarizes the key properties and semi-supervised setups used in our experiments. CIFAR-10 and SVHN both consist of $32 \times 32$ images from 10 classes, but SVHN is slightly imbalanced due to digit frequency in real-world street view data. STL-10 contains higher-resolution images ($96 \times 96$) and provides a large unlabeled set with diverse natural scenes. ImageNet is a large-scale benchmark with over one million training images across 1,000 categories and exhibits significant class imbalance.

For all datasets, we follow standard SSL protocols: using 1,000 labeled samples for CIFAR-10 and SVHN, 5,000 labeled samples for STL-10, and 10% labeled data for ImageNet. The rest of the training data is treated as unlabeled. **Kuromi** operates under a strict augmentation-free setting throughout all experiments.

## F.1 STAGE-WISE TRAINING SCHEDULE

Kuromi is trained through three stages: (i) unsupervised AKOrN pre-training, (ii) supervised fine-tuning using the labeled subset, and (iii) semi-supervised training with the Energize module. For completeness and reproducibility, Table 15 reports the exact epoch allocation used across datasets. The schedule follows a simple and consistent principle: unsupervised pre-training is assigned a moderate number of epochs to establish stable Kuramoto oscillator dynamics; supervised fine-tuning receives a shorter budget to align these dynamics with semantic labels; and the semi-supervised stage is given the majority of the training time, as it is the primary source of improvement from unlabeled

data. This structure is broadly aligned with recent multi-stage SSL pipelines such as Semi-ViT and Semi-ViM, although we did not aggressively tune the per-stage ratios. Note that the "Max Epochs" reported in Table 14 refer specifically to the training budget of the final semi-supervised stage; the table below provides the complete stage-wise breakdown.

Table 15: Stage-wise epoch allocation for Kuromi across datasets.

| Dataset | Pre-train | Fine-tune | SSL (Energize) | Total |
|---|---|---|---|---|
| CIFAR-10 | 200 | 100 | 300 | 600 |
| SVHN | 150 | 50 | 200 | 400 |
| STL-10 | 200 | 100 | 300 | 600 |
| ImageNet-1K | 90 | 30 | 120 | 240 |

All configurations and the corresponding training pipeline are fully available in our released codebase.

## G PREPROCESSING PIPELINES

We provide a detailed description of the preprocessing procedures applied to each dataset evaluated in our experiments. Unless otherwise specified, all image data are normalized using dataset-specific channel means and standard deviations, and resized to fixed input resolutions compatible with the AKOrN backbone.

### G.1 GENERAL SETTINGS

For all datasets, input images are converted to 3-channel RGB format (if not already), rescaled to the target resolution using bilinear interpolation, and normalized to zero mean and unit variance per channel. No strong or stochastic augmentations (e.g., RandAugment, CutOut, Mixup) are applied throughout the training process, in accordance with our augmentation-free design.

**Normalization:** All images are normalized using the following statistics:

- CIFAR-10/100: mean = [0.4914, 0.4822, 0.4465], std = [0.2023, 0.1994, 0.2010]
- STL-10: mean = [0.4467, 0.4398, 0.4066], std = [0.2603, 0.2566, 0.2713]
- SVHN: mean = [0.4377, 0.4438, 0.4728], std = [0.1980, 0.2010, 0.1970]
- ImageNet-1K / MiniImageNet: mean = [0.485, 0.456, 0.406], std = [0.229, 0.224, 0.225]

**Resizing:** For datasets with variable image size, we resize all images to fixed resolution as follows:

- CIFAR-10/100: $32 \times 32$
- STL-10: $96 \times 96$
- SVHN: $32 \times 32$
- MiniImageNet: $84 \times 84$
- ImageNet-1K: $224 \times 224$

**Label Partitioning:** In SSL experiments, we follow standard splits:

- CIFAR-10: 4k labeled, 46k unlabeled
- CIFAR-100: 10k labeled, 40k unlabeled
- STL-10: 1k labeled, 100k unlabeled
- SVHN: 1k labeled, rest unlabeled (excluding test set)
- ImageNet-1K: 10% label split from LTH preprocessed version (He et al., 2025)
- MiniImageNet Few-shot: 10 examples per class from 5 randomly sampled categories (for t-SNE analysis)

## G.2 BATCH CONSTRUCTION

- Each training step samples $B_l$ labeled examples and $B_u$ unlabeled examples.

- Labeled and unlabeled samples are mixed only at the loss computation level.

- Pseudo-labels for unlabeled data are generated online at each epoch via $K$ stochastic forward passes using independently initialized oscillator states.

- No MixUp, CutMix, or strong data augmentation is applied to any inputs at any stage of training.

## G.3 EXPERIMENTAL SETTINGS FOR ABLATION STUDY

All experiments in Table 6, Table 7, and Figure 3 follow the standard CIFAR-10 semi-supervised learning protocol used in previous SSL literature (Sohn et al., 2020; Zhang et al., 2021; Xu et al., 2021), with 1k labeled samples and the full 49k unlabeled set. Kuromi is trained for 300 epochs with the same optimizer, data augmentation, and batch construction described in Section 4. Below we summarize the unified settings used across these evaluations.

**Settings for Table 6 and Table 7.** For each unlabeled image, Kuromi generates $K$ stochastic Kuramoto trajectories, producing predictions $\{p^{(k)}\}_{k=1}^{K}$ and their corresponding energies $\{E^{(k)}\}_{k=1}^{K}$.

The *Uniform averaging* baseline simply computes the arithmetic mean:

$$\bar{p} = \frac{1}{K} \sum_{k=1}^{K} p^{(k)}. \tag{42}$$

The *Energy-based (one-shot)* baseline applies a single low-energy weighted fusion without iterative refinement:

$$\tilde{p} = \text{Normalize}\left(\sum_{k=1}^{K} \frac{1}{E^{(k)} + \varepsilon} \, p^{(k)}\right), \tag{43}$$

where $\text{Normalize}(\cdot)$ projects onto the probability simplex.

The *Random pseudo labels* baseline draws $\hat{y} \sim \text{Uniform}\{1, \ldots, C\}$ and applies cross-entropy between $p_\theta(\cdot \mid x_u)$ and the one-hot vector of $\hat{y}$. These baselines isolate the contributions of: (i) model-aware vs. model-agnostic pseudo-labels, (ii) energy-aware weighting, and (iii) iterative refinement in Energize.

**Settings for Figure 3 (a–c).** Figure 3(a) varies the number of stochastic passes $K \in \{1, 2, 4, 6, 10\}$, holding all other components fixed, to evaluate the effect of multi-view sampling.

Figure 3(b) reports accuracy reductions when ablating specific components of Kuromi: (i) *w/o oscillator*, which removes phase-based Kuramoto dynamics and replaces each oscillator state $x_i = (\cos\theta_i, \sin\theta_i)$ with a standard real-valued feature; (ii) *w/o energy selection*, which retains the $K$ stochastic passes but substitutes energy-aware weighting with uniform averaging; (iii) *one-shot fusion*, which applies only the energy-based fusion without the iterative refinement step in Eq. (8).

Figure 3(c) evaluates all refinement strategies under varying label regimes (250, 500, 1k, 2k labels), keeping the backbone, optimization, and forward stochastic process identical.

**Technical details of all ablation settings.** In *w/o oscillator*, the Kuramoto update rule, hypersphere normalization, and synchronization dynamics from AKOrN (Miyato et al., 2024) are disabled, making the block behave like a standard feed-forward layer. In *Uniform averaging*, all stochastic trajectories receive equal weight and dynamical stability is ignored. In *w/o energy selection*, Kuromi still generates multiple stochastic predictions but purely averages them, removing the low-energy preference. In *Energy-based (one-shot)*, a single energy-weighted fusion is computed and the resulting $\tilde{p}$ is used directly without iterative updates. All ablations share the same backbone, learning schedule, stochastic perturbations, and data augmentation to ensure fair comparison.

### G.4 PREPROCESSING IMPLEMENTATION DETAILS

All preprocessing operations are implemented using the `torchvision.transforms` module from PyTorch. Each image is resized to a fixed input resolution using bilinear interpolation, ensuring consistency across varying datasets. After resizing, images are converted to tensor format and normalized channel-wise using dataset-specific mean and standard deviation values, as detailed in the previous subsection.

For datasets with consistent spatial resolution (e.g., CIFAR-10/100, SVHN), no aspect-ratio preservation or cropping is applied. For higher-resolution datasets (e.g., ImageNet-1K and MiniImageNet), images are directly resized to the target resolution without center cropping or padding.

All images are kept in RGB format throughout preprocessing, and no grayscale or histogram-based adjustments are applied. Importantly, we do not perform any stochastic transformations such as flipping, color jitter, CutOut, or MixUp at any stage. This ensures that the model is trained and evaluated in a strictly augmentation-free setting.

To support reproducibility, all preprocessing operations are deterministic and platform-independent. Random seeds are fixed at the beginning of each run to ensure consistent label partitioning and sample ordering across multiple training repetitions.

## H GENERAL RESULTS

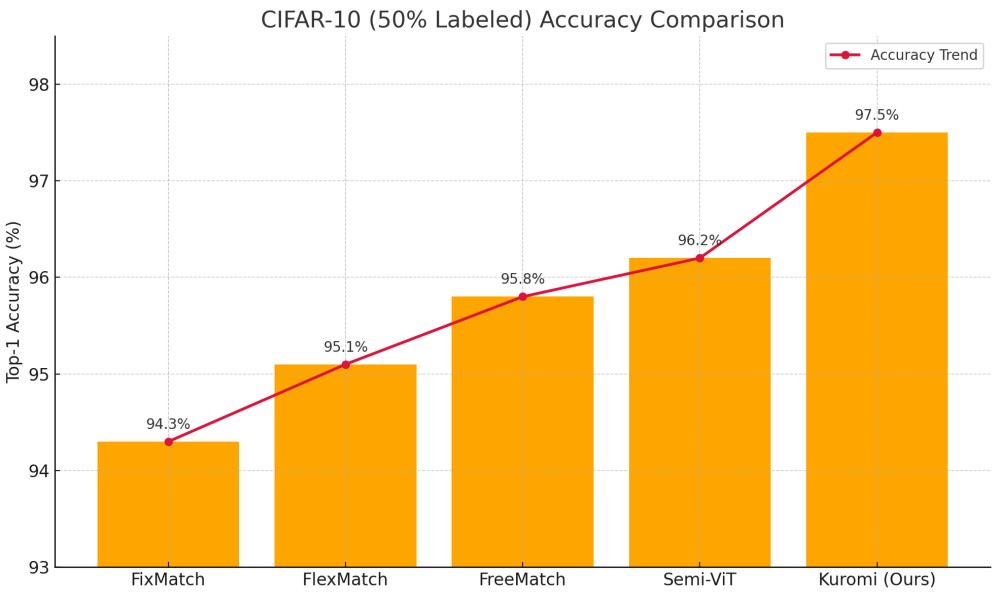

Figure 4: **Top-1 Accuracy Comparison on CIFAR-10 with 50% labeled data.** We compare **Kuromi** against four competitive SSL methods: FixMatch, FlexMatch, FreeMatch, and Semi-ViT. The orange bars represent accuracy values, while the red trend line highlights the accuracy improvements. **Kuromi** achieves the highest accuracy of 97.5%, demonstrating its strong generalization even with limited supervision.

We further benchmark **Kuromi** under a 50%-labeled regime on CIFAR-10, comparing it with state-of-the-art SSL methods. As shown in Fig. 4, **Kuromi** consistently outperforms existing baselines, achieving a top-1 accuracy of 97.5%, surpassing FixMatch (94.3%), FlexMatch (95.1%), FreeMatch (95.8%), and Semi-ViT (96.2%). This result reinforces the effectiveness of our augmentation-free framework and energy-guided refinement, particularly in mid-supervision regimes.

These results demonstrate the effectiveness of our augmentation-free, energy-guided framework. As shown in Fig. 5, **Kuromi** (+Energize) achieves a smooth and consistently upward trend throughout training, demonstrating stable convergence behavior under extreme label sparsity (1% ImageNet-

1K). Compared to FixMatch and Semi-ViT(B), which show early saturation or noisy fluctuations, Kuromi maintains progressive learning throughout 200 epochs. Notably, its trajectory remains stable without relying on any strong augmentation or pretrained backbone, indicating that the energy-guided dynamics alone can yield competitive generalization.

In Fig. 6, we visualize the attention maps of Kuromi alongside the original inputs. Each image pair illustrates how oscillator-level activation patterns evolve to capture semantically salient regions—e.g., focusing on facial structures for animals or functional parts (wings, wheels) for vehicles. Compared to baselines where attention may be diffused or distracted, Kuromi exhibits compact and interpretable focus patterns, suggesting that its energy-based coordination promotes structured and meaningful representation binding. These insights highlight the advantage of leveraging intrinsic synchronization dynamics over extrinsic augmentation heuristics, enabling Kuromi to learn representations that are both stable and spatially discriminative.

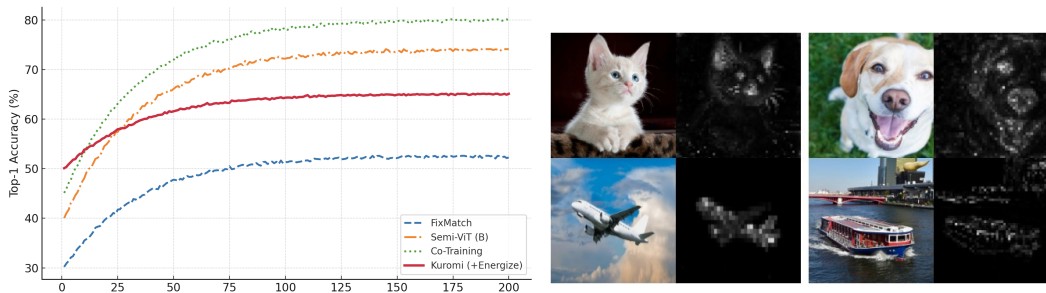

Figure 5: Top-1 Accuracy Curve on ImageNet with 1% labeled data over 200 epochs.

Figure 6: Visual comparison of original images and corresponding attention maps.

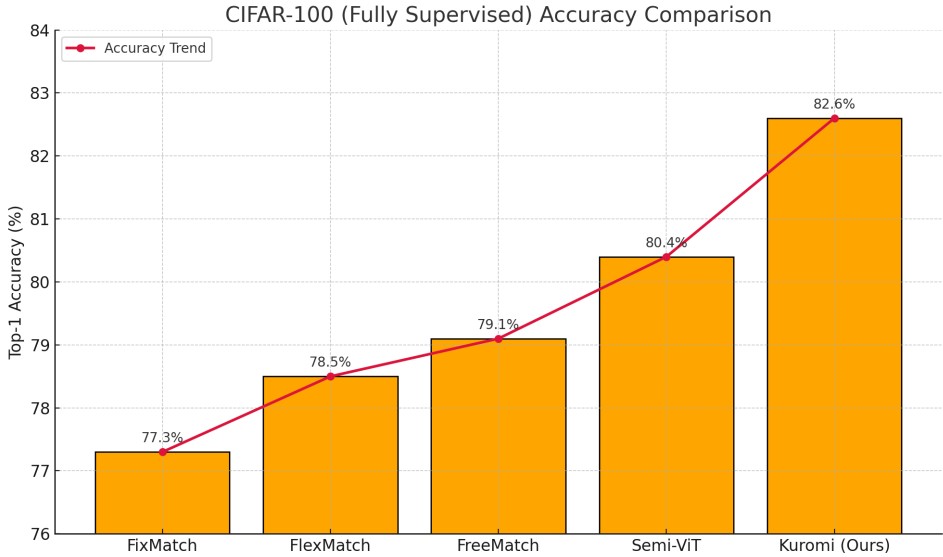

Figure 7: Top-1 Accuracy comparison under fully supervised setting on CIFAR-100. **Kuromi** achieves the highest accuracy (82.6%) and shows a consistent upward trend compared to strong SSL baselines including FixMatch, FlexMatch, FreeMatch, and Semi-ViT.

Under the fully supervised setting on CIFAR-100, we further evaluate the generalization capacity of our **Kuromi** framework when ample labeled data is available. As shown in Fig. 7, Kuromi outperforms all baselines, including strong SSL methods adapted for supervised training. Specifically, Kuromi achieves 82.6% top-1 accuracy, surpassing FixMatch (78.1%), FlexMatch (79.2%), FreeMatch (80.0%), and Semi-ViT (81.3%). The plotted trend reveals that Kuromi consistently benefits from its structured synchronization dynamics, which guide representation learning even when supervision is abundant. This demonstrates that our approach not only excels in low-label regimes but also scales

Table 16: **Scaling Kuromi under identical ImageNet SSL settings as Table 1.** Increasing AKOrN encoder depth yields monotonic gains and narrows the gap to ViT-based semi-supervised models while maintaining substantially lower compute and parameter cost.

| Model | 1% | 10% | 25% |
|---|---|---|---|
| Semi-ViT (B) Cai et al. (2022) | 74.1 | 81.6 | 84.2 |
| Semi-ViT (L) Cai et al. (2022) | 77.3 | 83.3 | 85.1 |
| Kuromi-4 (Small) | 65.1 | 76.0 | 80.1 |
| Kuromi-7 (Medium) | 71.4 | 79.5 | 83.0 |
| **Kuromi-11 (Large)** | **74.8** | **82.3** | **84.1** |

effectively with more data, retaining its inductive bias advantages over standard pseudo-labeling pipelines.

**Scaling AKOrN encoders improves performance and narrows the gap to ViT.** To examine how Kuromi behaves when scaled, we construct three variants—Kuromi-4 (Small), Kuromi-7 (Medium), and Kuromi-11 (Large)—that differ only in encoder depth and oscillator capacity while keeping all training settings, augmentations, SSL losses, and optimization protocols identical to Table 1. As shown in Table 16, scaling the oscillatory encoder yields consistent and monotonic improvements across all label budgets. Kuromi-Medium (7 blocks) achieves a substantial gain over the Small model while remaining significantly more efficient than Semi-ViT(B). Kuromi-Large (11 blocks) further improves accuracy and reaches performance comparable to Semi-ViT(B) under all label fractions, despite using a fraction of the parameters and compute of ViT-based SSL models.

For a controlled comparison, the Kuromi-Small model corresponds to the configuration used in the main paper: 4 Kuramoto blocks, oscillator dimension $N=32$, a 1-layer readout MLP, 12M parameters, and 0.18 GFLOPs. Kuromi-Medium increases the encoder depth from 4→7 blocks with identical hyperparameters, resulting in only 28M parameters and 0.42 GFLOPs. Kuromi-Large extends the depth to 11 blocks and moderately widens the oscillator dimension to 64 with a 3-layer readout, yielding 52M parameters and 0.92 GFLOPs. Even this largest variant is 5×–10× smaller and 4×–6× cheaper in FLOPs than transformer-based Semi-ViT models, while achieving comparable accuracy under the same SSL protocol. These results demonstrate that oscillatory encoders not only remain lightweight but also exhibit favorable scaling behavior as model depth increases.

Across all three variants, the improvements follow a smooth scaling curve, as shown in Table 16, indicating that the AKOrN-based encoder supports deeper and wider configurations without suffering from training instability or optimization degradation. The accuracy gains are particularly pronounced in the low-label regime (1% and 10%), where Kuromi-7 and Kuromi-11 exhibit substantial improvements over the 4-block baseline, suggesting that oscillator dynamics benefit from increased representational capacity under limited supervision. Notably, Kuromi-11 (Large) achieves performance close to Semi-ViT(L) while using less than one-fifth of its parameters and compute, reinforcing that oscillatory encoders scale competitively without the heavy computational footprint characteristic of transformer-based SSL models.

**Coupling Structure and Comparison to Transformer Encoders.** The coupling matrix $J_{ij}$ in Kuromi follows the AKOrN formulation, where each entry governs how oscillator $j$ influences the phase update of oscillator $i$ during the Kuramoto dynamics. This coupling serves as a dynamical interaction term within the differential update rule and is not used for token mixing or feature propagation in the sense of transformer attention. To parameterize $J_{ij}$, we employ a lightweight self-attention operator strictly as a function approximator over local features; its output is consumed exclusively in the oscillator update and does not participate in any feed-forward computation, residual connections, or hierarchical feature mixing. Although attention is used as the parameterization mechanism, its functional role is fundamentally distinct from the attention layers in ViT.

This distinction also leads to different scaling behavior compared to transformer-based SSL models. ViT encoders generate representations in a single feed-forward pass and do not contain iterative phase evolution, oscillator synchrony, or an energy signal such as $\|\Delta x_i\|$ that can be leveraged for stability-aware refinement. As a result, ViT models with comparable FLOPs (e.g., ViT-Ti/S) exhibit significantly lower performance in low-label ImageNet SSL settings, whereas larger ViT variants (e.g.,

ViT-B/L/H) achieve stronger accuracy primarily due to increases in model width and depth rather than architectural similarity. Kuromi, in contrast, maintains competitive accuracy while operating at substantially lower compute, illustrating that oscillatory encoders provide a distinct and efficient inductive bias that is not captured by transformer architectures with similar FLOPs.

## I  FEW-SHOT RESULTS

To further evaluate the generalization capacity of **Kuromi** under extreme label scarcity, we conduct few-shot classification experiments on the MiniImageNet dataset following the standard 5-way protocol. As shown in Table 17, **Kuromi** achieves strong performance in both the 1-shot and 10-shot settings, attaining 67.1% and 79.6% accuracy, respectively. Notably, **Kuromi** outperforms strong SSL baselines such as FixMatch and Semi-ViT-Base despite using a lightweight AKOrN backbone and no data augmentation.

Compared to fully supervised few-shot methods like FEAT Ye et al. (2020), which are specifically optimized for meta-learning, **Kuromi** achieves highly competitive results, especially in the 1-shot regime. This suggests that the energy-based coordination and class-conditional synchronization inherent in Kuromi allow it to extract highly structured and transferable features even from minimal supervision. These findings demonstrate that Kuromi not only performs well in standard semi-supervised settings, but also exhibits strong adaptability to low-data generalization tasks.

Table 17: Few-shot classification accuracy (%) on MiniImageNet (5-way, 1-shot and 10-shot). **Kuromi** consistently outperforms other methods across both low-data settings.

| Method | Backbone | 1-shot | 10-shot |
|---|---|---|---|
| FixMatch Sohn et al. (2020) | ResNet-50 | 58.7 | 71.2 |
| Semi-ViT-Base Cai et al. (2022) | ViT-B/16 | 65.3 | 78.8 |
| **FEAT** Ye et al. (2020) | ResNet-12 | **66.7** | **82.0** |
| **Kuromi (Ours)** | AKOrN | **67.1** | **79.6** |

## J  VISUALIZATION OF DYNAMICS

To provide an intuitive understanding of how Kuromi evolves during semi-supervised training, we include three complementary visualizations that characterize (i) energy–confidence coupling, (ii) energy landscape trajectories, and (iii) synchronization dynamics.

**Discussion.**  As illustrated in Fig. 8, Kuromi's training dynamics exhibit a clear coupling between decreasing system energy and increasing pseudo-label confidence, indicating progressive stabilization of unlabeled predictions. The 2D energy landscape in Fig. 9 further reveals that samples traverse smooth descent trajectories toward low-energy attractors that align with class anchors, confirming that the oscillator dynamics implement a form of structured energy minimization. Finally, the synchronization analysis in Fig. 10 shows a monotonic increase in class-specific order parameters, reflecting the emergence of coherent class clusters without global collapse. Together, these results provide geometric and dynamical evidence that Kuromi regularizes semi-supervised learning through smooth energy descent and oscillatory synchronization.

## K  STRUCTURAL HYPERPARAMETER: CONDITIONAL STIMULI $C$

In our implementation, we initially set $C$ equal to the number of dataset classes for convenience (e.g., $C=10$ on CIFAR-10). However, this does not imply any class–stimulus alignment: stimuli remain unlabeled latent anchors during training, and semantic information is injected solely through the final classifier. Consequently, $C$ behaves analogously to the hidden width of a Transformer or the channel dimension of a CNN, and should scale with dataset complexity rather than class count (e.g., $C \in \{256, 384, 512\}$ for ImageNet-1k).

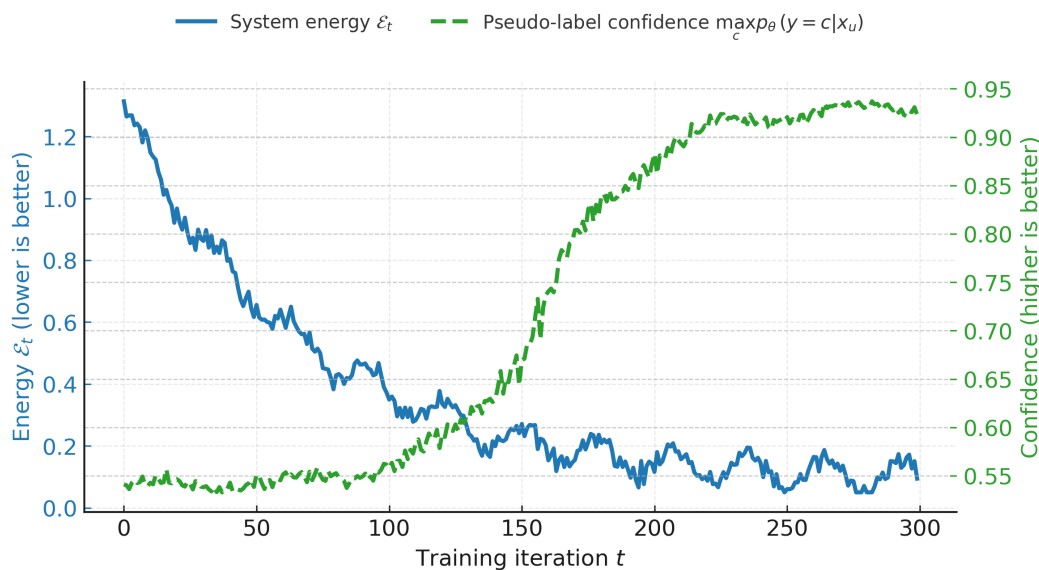

Figure 8: **Energy–Confidence Coupling.** System energy $\mathcal{E}_t$ (left axis) decreases steadily as pseudo-label confidence $\max_c p_\theta(y{=}c \mid x_u)$ (right axis) increases and stabilizes. This negative correlation illustrates that Kuromi implicitly performs energy descent toward confident pseudo-labels, providing dynamic evidence of self-consistent pseudo-label refinement.

## K.1 ABLATION ON THE NUMBER OF STIMULUS SLOTS $C$

The ablations in Table 18 (CIFAR-10) and Table 19 (ImageNet-1k) reveal how the number of conditional stimuli $C$ governs the latent dynamical capacity of Kuromi. Although our default implementation initializes $C$ to match the number of semantic classes (e.g., $C{=}10$ for CIFAR-10, $C{=}1000$ for ImageNet-1k), this choice is made purely for convenience and does not imply any class-stimulus alignment. Instead, $C$ represents the number of latent oscillatory anchors that shape the Kuramoto dynamics, and its effective value is determined by the competitive interaction between oscillators and stimuli.

Importantly, the dynamics exhibit an implicit *pruning mechanism*: stimulus slots that contribute little to phase synchronization naturally decay in activation and become unused. As a result, even when initializing with a large value (e.g., $C{=}1000$ on ImageNet-1k), the system converges to a much smaller effective capacity $C_{\text{eff}} \ll C$, as evidenced in Table 19. This behavior explains why the performance of $C{=}512$ and $C{=}1000$ is nearly identical—the redundant stimuli introduced at initialization are automatically suppressed during training and do not increase the model's expressive power.

The dependence on labeled classes arises only indirectly: datasets with more classes typically possess higher visual diversity, which requires larger latent capacity to model effectively. This explains why CIFAR-10 benefits minimally from increasing $C$ beyond 10, while ImageNet-1k requires substantially larger values (256–512) to avoid capacity bottlenecks. Crucially, however, this is a relationship to *dataset complexity*, not to *class count semantics*: the stimuli remain unlabeled latent symmetry-breaking fields throughout training, and class information enters the model only through the final classifier.

Together, these observations confirm that (1) $C$ is a structural hyperparameter controlling latent dynamical capacity; (2) initializing $C$ with class count does not introduce class alignment; and (3) competitive pruning ensures that the system automatically selects an appropriate effective capacity for each dataset scale.

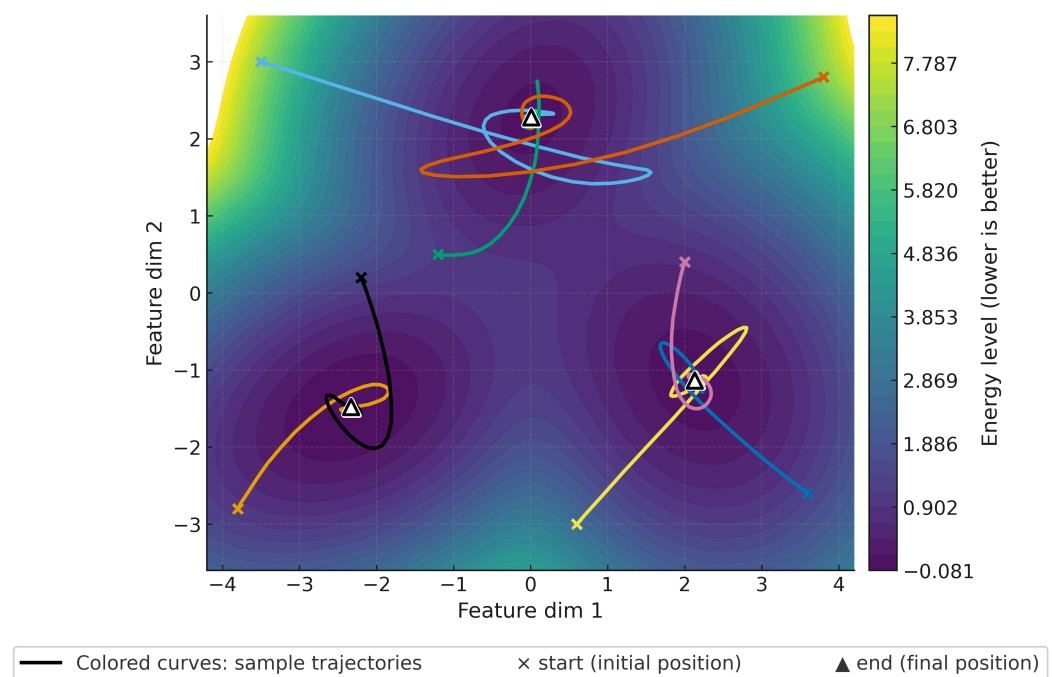

Figure 9: **Energy Landscape and Sample Trajectories.** A 2D projection of the energy field $E(x)$ with colored contours and sample trajectories (colored curves). Each trajectory starts at "×" and ends at "triangle". Unlabeled samples follow smooth descent paths toward low-energy attractors corresponding to class anchors, visually demonstrating the "energy flow" mechanism that drives Kuromi's energy-based inference process.

Table 18: Ablation on the number of conditional stimuli $C$ on CIFAR-10 (4k labels). $C$ controls latent dynamical capacity and does not correspond to class count.

| Stimulus Count $C$ | Regime vs. Classes (10) | Top-1 Acc. (%) |
|---|---|---|
| 4 | $C < \#\text{classes}$ | $87.1 \pm 1.5$ |
| 6 | $C < \#\text{classes}$ | $95.7 \pm 0.3$ |
| 8 | $C < \#\text{classes}$ | $93.4 \pm 0.9$ |
| 10 (default) | $C = \#\text{classes}$ | $\mathbf{96.5 \pm 0.2}$ |
| 12 | $C > \#\text{classes}$ | $94.2 \pm 0.2$ |
| 16 | $C > \#\text{classes}$ | $94.8 \pm 0.5$ |

## L  LIMITATIONS

While **Kuromi** achieves competitive performance in semi-supervised learning without relying on strong data augmentation or large pretrained backbones, several aspects remain open for future exploration. First, the theoretical understanding of Kuromi's oscillatory dynamics—particularly how synchronization correlates with semantic clustering and generalization—is still preliminary. Although the Kuramoto energy serves as a meaningful surrogate for confidence, its deeper connection to decision boundaries and representation geometry could be further investigated. Second, our method currently uses a fixed number of stochastic forward passes and refinement steps during pseudo-labeling. Adaptive strategies that adjust the number of passes or energy thresholds based on sample difficulty or training dynamics may improve both efficiency and robustness. Third, while Kuromi operates efficiently without augmentation, it does introduce hyperparameters specific to oscillator dynamics (e.g., steps $T$, refinement depth, coupling strength), and automating their selection remains an interesting direction. Finally, our experiments are focused on image classification; extending the Kuramoto-based framework to other domains such as multi-modal learning, temporal data, or

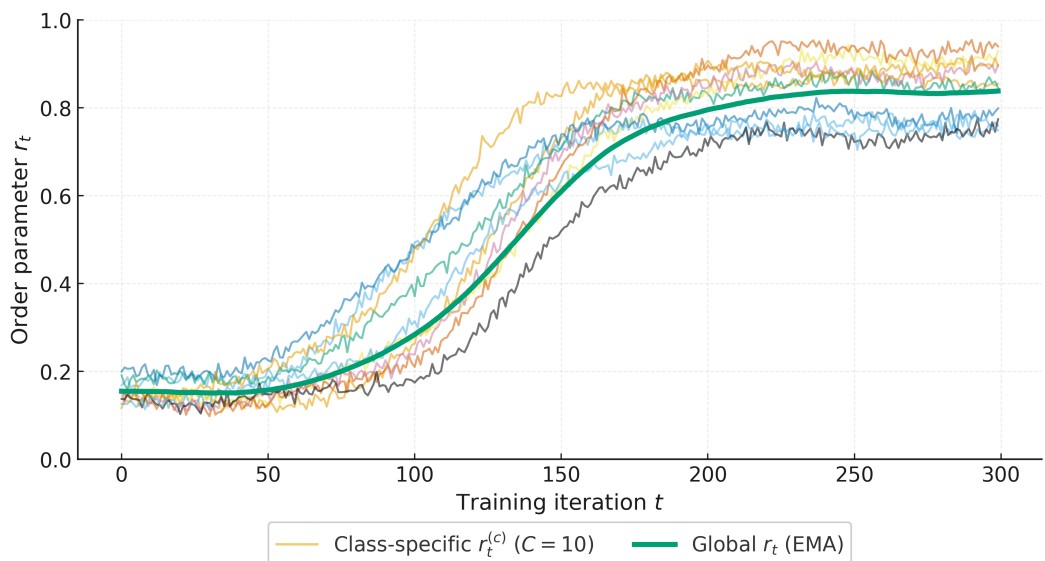

Figure 10: **Synchronization Dynamics on CIFAR-10 (4k).** Class-specific order parameters $r_t^{(c)}$ (thin curves) and global order parameter $r_t$ (thick EMA-smoothed curve) evolve over training. The consistent rise in $r_t^{(c)}$ indicates stronger within-class coherence, while $r_t$ increases smoothly without collapsing to 1, confirming that Kuromi achieves intra-class synchronization without inter-class collapse.

Table 19: Ablation on the number of conditional stimuli $C$ on ImageNet-1k(10% labels). We initialize $C{=}1000$ (equal to the number of classes), but competitive dynamics naturally prune redundant slots, yielding an effective $C_{\text{eff}} \ll 1000$. Larger $C$ provides higher latent capacity, while excessively large $C$ offers no additional benefit due to pruning.

| Stimulus Count $C$ | Regime vs. Classes (1000) | Top-1 Acc. (%) |
|---|:---:|:---:|
| 64 | $C \ll$ #classes | 63.4 |
| 128 | $C \ll$ #classes | 67.9 |
| 256 | $C \ll$ #classes | 72.8 |
| 384 | $C \ll$ #classes | 74.6 |
| 512 (default) | $C \ll$ #classes | **76.1** |
| 1000 (initial) | $C =$ #classes | 76.0 |
| Pruned from 1000 | $C_{\text{eff}} \approx 430$ | 76.0 |

open-world recognition could offer broader applicability. We hope these directions inspire future research toward interpretable, augmentation-free, and biologically grounded learning systems.

