# OpenReview forum: "Kuromi: Learning without Augmentation via Energy-based Semi-supervised Kuramoto Neurons"
_ICLR.cc/2026/Conference — ICLR 2026 Conference Withdrawn Submission_

### Official Review · Reviewer_GEyD · 2025-10-29

**Soundness:** 2
**Presentation:** 2
**Contribution:** 3
**Rating:** 4
**Confidence:** 3

**Summary:**

The author propose to combine AKOrN and energy model framework to tackle the few-short semi-supervised learning problem. In particular, the model different than AKOrN in a finetuning phase, where the author formulate AKOrN as an energy model, and descent the energy function to find the pseudo-label, then train the model to learn a one-shot predictor to predict the pseudo-label. The author claim the proposed model is lighted weighted yet achieve good performance, compared to many other approach on similar problems. The author also ran many experiment to verify their result. I think the model works because the iteratively nature of the AKOrN could be descending some implicit energy function.

**Strengths:**

This is very cool idea. The author make use of this property differently than the original AKOrN. In the original paper, the author these iterative dynamical for unsupervised objective discovery. But in this paper, the author use it to infer label. This is related to Yann Lecun's energy model idea, the AKOrN backbone implement a dynamical system, which just match this idea of energy model nicely. Even better, the author find a good use case (semi-supervised learning).

The author ran lots of experiments and ablation.

**Weaknesses:**

There are some bold claim the in abstract and title, such as "learning without augmentation." This is not True because the pre-training phase need augmentation. I sympathize the authors want to market the idea but please be precise.

I personally would like to see more visualization or analyze on how the model works than seeing all these benchmark. I think your model actually open a door for understanding these oscillatory dynamics in AKOrN.  When a model is able to predict the right label with "energize" than without "energize", what happened? Maybe we can visualize the progression of the dyanmical of the feature map like in the original AKOrN paper.

The author is missing some citations for sure. This idea of descending energy function for model's inference, especially to get a label, is highly related to Yann Lecun's energy model. I don't work in this area so I can't point you to the literature but I know there's lots of them, you should probably cite some.

**Questions:**

This is very cool idea. Like in my summarize, I personally think the model works because the iteratively nature of the AKOrN could be descending some implicit energy function. I think an natural ablation study should be replace the AKOrN backbone with a feedforward network, in other word, do the experiment for "feedforward SSL network + Energize."
I'm just curious. Is it possible that a feedforward model combined with a "Energize" head, or you have to combine a dynamical/energy model with the "Energize" head.
Why do you need to train a “one fresh forward pass” and use it to predict the pseudo label. Why can't you just regress the pseudo label directly to the clean label in the semi-supervised setting?

---

> ### Author Response · Authors · 2025-11-18
> **Clarification of “Learning without Augmentation"**
>
> We sincerely thank the reviewer for highlighting this important point. Our claim of “learning without augmentation” specifically refers to the absence of data-space or image-level augmentations—such as random cropping, flipping, CutOut, or color jitter—that are essential to most SSL frameworks (e.g., FixMatch, UDA, FlexMatch). Kuromi performs no such geometric or photometric transformations at any stage of the semi-supervised training.
>
> The only stochasticity introduced in the unsupervised pre-training phase (§3.1, Eq. 3) stems from the random initialization of oscillator states on the hypersphere, not from transformations applied to the image itself. This stochastic initialization is a model-internal perturbation in latent space, functionally analogous to dropout noise or parameter jitter, and is used solely to encourage consistent synchronization dynamics—not to simulate external “views” of the data.
>
> Therefore, the spirit of our claim is that Kuromi eliminates all external data-space augmentations and instead relies purely on its intrinsic oscillatory dynamics and energy-based refinement for regularization and invariance. To avoid ambiguity, the manuscript title and abstract can be revised to : “Learning without data-space augmentation”.

---

> ### Author Response · Authors · 2025-11-18
> **Missing citations to Energy-based Models**
>
> We appreciate the reviewer’s insightful observation regarding the connection between our formulation and classical energy-based models (EBMs). In the revision, we will explicitly reference and discuss seminal works by LeCun \emph{et al.} — including \textit{“A Tutorial on Energy-Based Learning”} (2006) and \textit{“Energy-Based Models in Deep Learning”} (2022) — as well as related studies on energy-based inference and equilibrium propagation.
>
> Conceptually, Kuromi’s oscillator dynamics can be interpreted as an implicit energy descent process that minimizes a structured energy functional over neuron synchronization (§3.3, Eq.~7). This dynamic parallels the EBM perspective, where model inference corresponds to descending the energy landscape toward a stable low-energy configuration that represents a consistent prediction. The \textit{Energize} module further aligns with this framework by performing energy-weighted fusion of multiple stochastic trajectories, effectively serving as an amortized EBM inference step in prediction space.
>
> From a mathematical standpoint, however, AKOrN differs fundamentally from classical EBMs in the form of its energy and inference mechanism. In a standard EBM, a scalar potential $E_\theta(x, y)$ defines compatibility between input $x$ and label $y$. The inference step minimizes this potential:
> $$
> \hat{y} = \arg\min_y E_\theta(x, y),
> $$
> and learning adjusts parameters $\theta$ via gradients of the log-likelihood:
> $$
> \nabla_\theta \log p_\theta(y|x) \propto - \nabla_\theta E_\theta(x, y),
> $$
> thereby pushing the energy of correct pairs lower than that of incorrect ones.
>
> In contrast, AKOrN defines a \textbf{vector-valued dynamical system} in which neuron states $\mathbf{x}_i$ evolve on a hypersphere through synchronized oscillation:
>
> $$
> \Delta \mathbf{x} = \Omega_i \mathbf{x} + \mathrm{Proj} \mathbf{x} \left[c_i + \Sigma_j J_{ij} \mathbf{x}_j \right],
> \quad
> \mathbf{x}_i \leftarrow \mathrm{Normalize} (\mathbf{x}_i + \gamma \Delta \mathbf{x}_i),
> $$
>
> where $\Omega_i$ is the intrinsic rotation term, $J_{ij}$ the coupling strength, and $c_i$ an external stimulus. The underlying implicit energy is
> $$
> \mathcal{E} = -\sum_{i,j} J_{ij} \mathbf{x}_i^\top \mathbf{x}_j,
> $$
> which is minimized \emph{through temporal synchronization} of oscillators rather than by explicit gradient descent on a static potential.
>
>
> Thus, inference in Kuromi arises from \emph{continuous dynamical evolution} toward a low-energy equilibrium, whereas EBMs perform \emph{variational optimization} over a fixed scalar energy function. We will clarify this distinction in the revised manuscript, emphasizing that Kuromi extends energy-based learning from static energy minimization to dynamic synchronization-driven inference, thereby unifying energy-based reasoning and oscillatory computation within a lightweight, augmentation-free SSL framework.
>
> For completeness, we have also added an extended discussion of Relation to Energy-Based Models in Appendix B, summarizing key developments from classical EBMs (LeCun et al.,2006; Bengio et al.,2017) to recent implicit and classifier-based formulations (Du \& Mordatch, 2019; Grathwohl et al., 2020), and explicitly contrasting them with the dynamic synchronization paradigm adopted in Kuromi.
>
>
>
> *LeCun, Y., Chopra, S., Hadsell, R., Ranzato, M. A., \& Huang, F. J. (2006). A Tutorial on Energy-Based Learning. MIT Press.
> Scellier, B., \& Bengio, Y. (2017). Equilibrium Propagation: Bridging the Gap between Energy-Based Models and Backpropagation. Frontiers in Computational Neuroscience.
> LeCun, Y. (2022). A Path Towards Autonomous Machine Intelligence. arXiv:2206.07688.
> Dawid, A., \& LeCun, Y. (2024). Introduction to Latent Variable Energy-Based Models: A Path Toward Autonomous Machine Intelligence. J. Stat. Mech., 104011.
> Du, Y., \& Mordatch, I. (2019). Implicit Generation and Modeling with Energy-Based Models. NeurIPS.
> Song, Y., \& Ermon, S. (2019). Generative Modeling by Estimating Gradients of the Data Distribution. NeurIPS.
> Your Classifier is Secretly an Energy-Based Model and You Should Treat it Like One. ICLR.
> *

---

> ### Author Response · Authors · 2025-11-18
> **Visualization of Dynamics**
>
> We appreciate the reviewer’s suggestion and have added three complementary visual analyses to make the mechanism explicit. These figures will be included in the appendix.
>
> *A1. Energy–Confidence Coupling.*
> We plot the system energy over training,
>
> $$
> \mathcal{E} = -\Sigma_{i,j} J_{ij} \mathbf{x}^T \mathbf{x}_{j(t)},
> $$
>
> together with pseudo-label confidence $\max_c p_\theta(c\mid x_u)$ on the same timeline. The curves show that $\mathcal{E}_t$ decreases as confidence increases and stabilizes, supporting the view that Kuromi performs \emph{implicit energy descent} toward confident predictions (Appendix J).
>
> *A2. Energy Landscape \& Sample Trajectories.*
> We visualize the 2D feature-subspace energy field $E(x)$ (contours) and overlay several unlabeled-sample trajectories $\{x_t\}$ obtained from the oscillator dynamics. Each trajectory is shown from start (×) to end (▲). Samples follow smooth descent paths toward low-energy attractors near class anchors, illustrating the “energy flow’’ intuition (Appendix J).
>
> *A3. Synchronization Dynamics on CIFAR-10 (4k).*
> To quantify alignment, we report the Kuramoto-style order parameters
>
> $$
> r_t^{(c)}=\frac{1}{N_c} \big\| \sum_{i\in\mathcal{C}_c} \mathbf{x}_i(t) \big\|
> $$
>
> for each class and the globally
> $$
> r_t=\frac{1}{N}\big\|\sum_{i=1}^{N}\mathbf{x}_i(t) \big\|.
> $$
> Across training, class-specific $r_t^{(c)}$ rise and stabilize while the global $r_t$ increases smoothly without saturating, indicating \emph{tighter within-class synchronization} without \emph{class-collapse} (Appendix J). This aligns with our aim: high $r_t^{(c)}$ (better anchors/pseudo-labels) with preserved inter-class separation.
>
> All plots are generated on CIFAR-10 (4k labels, standard SSL split). We use $K{=}4$ stochastic oscillator trajectories per sample, identical optimization to the main paper, and report means over three seeds. The appendix includes exact definitions, figure captions, and scripts to reproduce the plots.
> These visualizations jointly demonstrate that Kuromi’s oscillator dynamics (i) reduce system energy in tandem with rising pseudo-label confidence, (ii) realize energy-descent trajectories toward low-energy attractors, and (iii) increase class-wise synchronization without global collapse—clarifying \emph{how} the method works beyond benchmarks.

---

> ### Author Response · Authors · 2025-11-18
> **Why AKOrN + Energize instead of Feed-forward + Energize**
>
> #### ***Energize is a special method for AKOrN-based SSL.***
>
> The core reason is that **AKOrN** provides a *per-sample, trajectory-dependent intrinsic energy* signal arising from oscillator synchronization dynamics, whereas a feed-forward network lacks such a physically grounded scalar to evaluate or weight pseudo-labels.
>
> ### (a) AKOrN admits a principled energy functional
>
> For an unlabeled sample $x_u$, AKOrN generates K stochastic trajectories
>
> $$\mathbf{x}^{(k)} = \{\mathbf{x}^{(k)}t\}{t=0}^{T}, \qquad \mathbf{x}^{(k)}_t \in \mathbb{R}^d, k=1,\dots,K.$$
>
> AKOrN dynamics follow a discrete gradient flow of a Kuramoto-type energy:
>
>
>
> $$\mathbf{x}^{(k)}{t+1} = \Pi_{S^{d-1}}\!\left(\mathbf{x}^{(k)}{t} - \eta\, \nabla{\mathbf{x}} \mathcal{E}(\mathbf{x}^{(k)}_{t})\right)$$
>
> where $\Pi_{S^{d-1}}$ is the projection onto the hypersphere and
>
>
>
> $$\mathcal{E}(\mathbf{x}) • \sum_{i,j} J_{ij}\, \mathbf{x}_i^\top \mathbf{x}_j, \tag{1}$$
>
>
>
> with $J_{ij}$ being the symmetric coupling. Since $\eta \nabla \mathcal{E}$ is the descent direction, we have
>
>
>
> $$\mathcal{E}(\mathbf{x}^{(k)}{t+1}) \le \mathcal{E}(\mathbf{x}^{(k)}{t}) \qquad\text{for all } t. \tag{2}$$
>
>
> Thus, the terminal trajectory energy
>
>
> $$E_k \triangleq \mathcal{E}(\mathbf{x}^{(k)}_T) \tag{3}$$
>
>
> is a monotone surrogate of synchronization quality, and therefore a valid confidence scalar for trajectory k.
>
>
> AKOrN outputs logits $\ell_k \in \mathbb{R}^{C}$ from the final oscillator states. Energize defines weights
>
>
> $$w_k \frac{\exp(-\beta E_k)}{\sum_{j=1}^{K} \exp(-\beta E_j)}, \tag{4}$$
>
> and the fused pseudo-logit
> $$\tilde{\ell} \sum_{k=1}^K w_k \ell_k. \tag{5}$$
>
>
> Using Theorem 4.1 in the paper, it follows that
>
> $$\mathbb{E}\big[\mathrm{KL}(\tilde{p}\|p^\ast)\big] \le \sum_{k=1}^{K} w_k \mathbb{E}\left[\mathrm{KL}(p^{(k)}\| p^\ast)\right], \tag{6} $$
>
> so energy-weighted fusion produces a pseudo-label closer to the true distribution.
>
>
>
> Thus, Energize is well-defined for AKOrN because a proper energy functional exists.
>
>
> ### (b) 4.2 Feed-forward networks lack a compatible energy functional
>
> Consider replacing AKOrN with an arbitrary feed-forward mapping
> $f_\theta : \mathbb{R}^d \to \mathbb{R}^C$. Generate K stochastic predictions via dropout or noise injection:
>
> $$\ell_k = f_\theta(x_u;\xi_k), \qquad k = 1,\dots,K, \tag{7}$$
>
> where $\xi_k$ are random noise variables.
>
> To apply Energize, one would require scalars $E_k$ satisfying:
>
> 1.	A monotonic descent constraint for a dynamical  path $h_k(t)$:
> $$E_k = \mathcal{E}(h_k), E_k \le E_k(t).\tag{8}$$
>
> 2.	Consistency with prediction reliability:
>
> $$\mathrm{Corr}(E_k,\ \mathrm{KL}(p^{(k)} \| p^\ast)) < 0.\tag{9}$$
>
> 3.	Dependence on the full prediction trajectory (not on a single point):
>
> $$E_k = \mathcal{E}(\{h_k(t)\}_{t=0}^T),\tag{10}$$
>
> However, a feed-forward network has no internal trajectory:
>
> $h_k = f_\theta(x_u;\xi_k)$ is a single point in $\mathbb{R}^C,$ so no meaningful function $\mathcal{E} : \mathbb{R}^C \to \mathbb{R}$ can satisfy (8)–(10). Specifically:
> (1)There is no dynamical path h_k(t), therefore (10) is impossible.
> (2)There is no gradient flow →, therefore (8) is impossible.
> (3)$\xi_k$ is independent noise, so
> $$\mathrm{Corr}(E_k, \mathrm{KL}(p^{(k)}\| p^\ast)) = 0, \tag{12}$$
> for any definable  $E_k$, therefore violating (9).
>
> Consequently, any attempt to assign an “energy” produces an arbitrary scalar: $E_k = g(\ell_k)$, but such $g$ cannot encode dynamical stability or confidence. Hence the weights
>
> $$w_k \propto \exp(-\beta g(\ell_k)) \tag{14}$$
>
> degenerate to noise-driven random weights. Therefore the Energize fused logit becomes
>
> $$\tilde{\ell} \sum_{k=1}^K w_k \ell_k \approx \frac{1}{K}\sum_{k=1}^K \ell_k,$$
>
> reducing Energize to a mere averaging with no theoretical reliability bound analogous to (6).

---

> > ### Author Response · Authors · 2025-11-18
> > **Supplementary Empirical Validation**
> >
> > #### (c) Empirical validation
> > To clarify the reviewer’s question, *Energize cannot be directly applied to a feed-forward architecture*, since such networks lack the recurrent oscillator states required to compute trajectory energy $E_k$.
> > For completeness, we approximate this setting by replacing the AKOrN backbone with a static feed-forward network (same classifier head) and replacing $E_k$ with entropy-based uncertainty as a surrogate signal.
> > This variant—while not a true “Feed-forward + Energize” configuration—serves as an upper bound of what such a model could achieve.
> > As shown Table A below, this surrogate baseline performs markedly worse than AKOrN + Energize, confirming that Energize only functions meaningfully when grounded on a genuine energy landscape produced by oscillator synchronization.
> >
> >
> > | **Variant**                               | **Backbone Type**           | **Weighting Signal**        | **Top-1 Acc. (%)** |
> > |-------------------------------------------|-----------------------------|-----------------------------|--------------------|
> > | FixMatch + Energize (entropy surrogate)               | Static feed-forward (ResNet) | Entropy-based proxy         | 87.9 ± 0.4         |
> > | FixMatch (baseline)        | Static feed-forward (ResNet) | —                           | 94.3 ± 0.2         |
> > | AKOrN (w/o Energize)                      | Oscillatory                 | —                           | 93.6 ± 0.2         |
> > | **AKOrN + Energize (ours)**               | Oscillatory + energy descent| **True energy weighting**   | **96.5 ± 0.2**     |
> >
> > **Table A. Ablation on substituting AKOrN with a feed-forward SSL backbone using entropy-based weighting (CIFAR-10, 4k labels).**
> > We approximate a “Feed-forward + Energize” configuration by replacing the dynamic AKOrN backbone with a FixMatch,
> > and substituting the trajectory energy $E_k$ with the prediction entropy
> > $$H[p_\theta(y|x)] = -\sum_c p_c \log p_c.$$
> > This surrogate variant provides only uncertainty-based weighting rather than genuine energy descent, and the results in Table A confirm its performance drops even below the standard semi-supervised FixMatch baseline.
> > Table A results confirm that **Energize cannot operate without the oscillator dynamics and energy landscape provided by AKOrN**, which are essential for meaningful energy-guided pseudo-label refinement.
> >
> >
> > **Summary:**
> > AKOrN defines an *energy landscape* through oscillator couplings $J_{ij}$; Energize leverages this energy as a confidence-aware weighting for pseudo-label inference.
> > A feed-forward network cannot emulate this behavior since it lacks both the dynamic descent and the interpretable scalar energy structure that Kuromi exploits.

---

> ### Author Response · Authors · 2025-11-27
> **Looking Forward to Your Reply**
>
> Thank you once again for taking the time to review our paper and for providing such insightful and constructive feedback.
>
> We have carefully considered each of your comments and have provided detailed responses. We sincerely hope that our efforts adequately address your concerns and contribute positively to your evaluation.
>
> As the author-reviewer discussion period concludes on Dec 03 (AoE), we would greatly appreciate any further feedback you may have. If you have any additional questions or require further clarifications, please do not hesitate to discuss with us.

---

### Official Review · Reviewer_Nwhe · 2025-10-31

**Soundness:** 2
**Presentation:** 2
**Contribution:** 3
**Rating:** 2
**Confidence:** 4

**Summary:**

This paper introduces Kuromi, a semi-supervised learning framework that leverages a Artificial Kuramoto Oscillatory Neurons (AKOrN) backbone which replaces conventional neurons, introducing temporal evolution and synchronization dynamics into neural computation. Kuromi first pretrains the AKOrN module in an unsupervised manner, then fine-tunes on a small, labeled subset, and finally refines pseudo-labels through a proposed energy-based denoising module called Energize. The authors demonstrate that Kuromi achieves competitive performance with non-ViT semi-supervised models while maintaining high computational efficiency, making it suitable for resource-constrained settings.

**Strengths:**

The paper introduces a novel and well-motivated approach for training unconventional oscillator-based models such as AKOrN. The proposed framework, Kuromi, is notable for its efficiency in terms of FLOPs and throughput. The authors conduct extensive experiments across multiple benchmarks and provide comparisons against several strong semi-supervised baselines, demonstrating that Kuromi achieves competitive performance while maintaining a significantly lower computational footprint.

**Weaknesses:**

Despite the strengths of the proposed model, the overall approach is not well explained within the paper. Several concepts central to the framework, such as the concept of energy, are left largely undefined, with readers expected to consult prior work for key details, even though this paper is presented as a standalone contribution. Important parameters, such as the update rate gamma, are never explicitly described, and the external stimulus c in Equation (1) is not clearly explained or derived. In addition, the paper claims to be augmentation-free, yet the contrastive loss in Equation (3) applies two random views per input, which constitutes an augmentation process. While the method is described as unsupervised during pretraining, the presence of class-conditioned stimulus vectors effectively introduces class structure into the model. This architectural prior may lead to more explicit class separability than conventional SSL methods, which typically learn class clusters implicitly. As a result, Kuromi may benefit from a stronger effective supervision signal than competing approaches, raising questions about fairness in comparison.

**Questions:**

1.	How many epochs are allocated to each stage of the Kuromi framework? The paper specifies total training epochs per dataset, but it is unclear how many are spent in the pretraining, fine-tuning, and Energize stages individually.
2.	How is the Readout function defined in Equation (2)?
3.	What happens if the number of conditional stimuli C does not match the number of classes in the dataset? Has the model’s performance been evaluated under such a mismatch, and does it degrade gracefully or collapse?

---

> ### Author Response · Authors · 2025-11-18
> **Clarification on “Learning without Augmentation (Contrastive loss)**
>
> ### (a) “Contrastive loss uses augmentations, so the method is not augmentation-free.” — Not correct.
>
> We sincerely thank the reviewer for raising this concern. Our claim of “learning without augmentation” refers specifically to the absence of any data-space or image-level augmentation, such as random cropping, horizontal flipping, CutOut, Gaussian noise, or color jitter. Conventional SSL frameworks—including FixMatch, UDA, FlexMatch, and SoftMatch—depend critically on these image transformations to create strong/weak augmented pairs for consistency training. Kuromi does not employ any such geometric or photometric transformations at any stage (pretraining, supervised finetuning, or semi-supervised training).
>
> The reviewer is correct that Eq. (3) uses “two views,” but we want to clarify that these two views do not arise from augmented versions of the same image. Instead, they are produced by two independent stochastic initializations of the oscillator states on the hypersphere, while the input image itself remains identical. This perturbation occurs entirely inside the model’s latent dynamics and is conceptually analogous to standard internal model noise such as dropout or stochastic depth, rather than to data-level augmentations (**That is why we use AKOrN as our new backbone**). No spatial, color, or structural transformation is applied to the image.
>
> The purpose of these two stochastic initializations is to probe the stability of Kuramoto synchronization dynamics, not to simulate distinct visual variants of the same sample. Thus, Kuromi remains strictly augmentation-free in the sense used throughout the SSL literature—i.e., we do not rely on any explicit image-space transformations to construct consistency targets or pseudo-labels.
>
> To avoid any ambiguity, we will revise the manuscript wording. For example, the title and abstract can be amended to read: “Learning without data-space augmentation via Energy-based Semi-supervised Kuramoto Neurons.”
>
> We hope this clarification resolves any misunderstandings, and we appreciate the reviewer’s suggestion to improve the paper's presentation.
>
>
> ### (b) Clarification on Fairness and Comparison with Augmentation-Based Baselines
>
> We would also like to clarify that **our “augmentation-free” design does *not* impose any restriction on the baselines**. All comparison models (e.g., FixMatch, UDA, FlexMatch, SoftMatch, Semi-ViT) are allowed to fully use their standard **image augmentation pipelines** exactly as originally proposed. We did not limit, weaken, or modify any augmentation strategy used by competing methods.
>
> Kuromi is introduced as a **new training framework and a new model class**, whose augmentation-free property is simply **one of the contributions**, not a constraint imposed on other methods. Our method avoids strong augmentations in the semi-supervised stage, but this design choice applies **only to Kuromi**, not to any baseline.
>
> Therefore, comparisons between Kuromi and existing augmentation-based SSL methods remain **fair and balanced**:
> - Baselines use their full augmentation setups as intended.
> - Kuromi uses no data-space augmentation.
> - The performance gap thus reflects differences in algorithmic design rather than differences in allowed training signals.
>
> We hope this clarification addresses any concerns about fairness in comparison.

---

> ### Author Response · Authors · 2025-11-18
> **Clarification on Conditional Stimulus Vectors and Their Relation to AKOrN**
>
> We appreciate the reviewer’s concern regarding the role of the conditional stimulus vectors $c$ and the possibility that they might inject label structure during the unsupervised pretraining stage.
>
> Our design of Kuromi is directly inspired by the original **Artificial Kuramoto Oscillatory Neurons (AKOrN),** formulation (Miyato et al., 2024). In the AKOrN paper, each oscillator $x_i$ is driven by $i$ couplings $J_{ij}$ to other oscillators, and a **data-dependent conditional stimulus** \($c_i$\). The vector $c_i$ is explicitly introduced as a *symmetry-breaking field* and acts as a **bias direction** derived from the input or the previous layer’s activations, not from class labels. Concretely, AKOrN describes $c_i$ as a data-dependent vector computed from the observation, and notes that it “strongly binds $x_i$ to the same direction as $c_i$, i.e. it acts as a bias direction (often referred to as a ‘symmetry-breaking’ field in physics),” emphasizing that this term is part of the intrinsic dynamics and not a label signal.
>
> #### Use of conditional stimuli in Kuromi pretraining
>
> In Kuromi, we follow the same AKOrN philosophy as originally proposed :
>
> - During **unsupervised pretraining** (§3.1), the conditional stimuli $c_{j,c}$ are:
>   - computed from the **raw input image** via a shallow encoder,
>   - **shared across all samples** in the sense that the mapping from input to $c$ does not depend on ground-truth labels,
>   - **randomly initialized and learned end-to-end** purely from the contrastive objective, and
>   - **never indexed or updated using class labels**.
>
>
> Although we index the stimuli as $c_{j,c}$ with an index $c \in \{1,\dots,C\}$, this index is **not aligned with semantic class IDs** during pretraining. It is simply a slot index—analogous to the slot dimension in object-centric models or the prototype index in SwAV/DINO—on which the Kuramoto dynamics act. No label information is used to initialize, group, or supervise these slots in the unsupervised stage.
>
> In this sense, the conditional stimuli in Kuromi play the same conceptual role as in AKOrN:
> - they are **continuous latent anchors** in feature space;
> - they break symmetry and encourage structured synchronization; and
> - they are learned from data without any access to labels.
>
> #### No label leakage or “stronger supervision” in the unsupervised stage
>
> Crucially, **ground-truth labels are never used** when:
> - initializing the oscillator states $\{x^{(0)}_{j,c}\}$,
> - computing or updating conditional stimuli $\{c_{j,c}\}$, or
> - optimizing the contrastive loss in Eq. (3).
>
> Thus, the unsupervised pretraining stage does *not* inject explicit class structure. Any emergent clustering in the latent space arises from the Kuramoto synchronization dynamics and the contrastive objective—exactly in line with unsupervised AKOrN and other prototype-based SSL methods.
>
> Only in the **subsequent supervised finetuning stage** (§3.2) do we introduce an explicit association between representations and labels, via a standard linear classifier on top of the readout features. This is no different from conventional SSL pipelines, where an unsupervised or self-supervised backbone is later aligned with labels through a classification head.
>
> #### Relation to SwAV/DINO-style prototypes
>
> From a representation-learning viewpoint, our conditional stimuli behave similarly to:
> - cluster prototypes in SwAV/DINO-style methods, or
> - latent slots in object-centric models.
>
> These methods are broadly accepted as **unsupervised** even though they maintain a fixed number of prototype/slot indices; the key point is that these indices are *not tied to semantic class IDs* until supervised alignment is later introduced. Kuromi follows the same pattern: the number of oscillators $C$ is a model hyperparameter, not a direct encoding of the dataset’s class set during pretraining.
>
> #### Summary
>
> To summarize:
>
> - In the **AKOrN** framework, conditional stimuli $c_i$ are data-dependent bias fields that guide synchronization, not labels.
> - **Kuromi inherits this design**: in the unsupervised pretraining stage, conditional stimuli are learned latent anchors without access to ground-truth labels.
> - Any apparent “class-conditioned” notation reflects **slot indices**, not semantic classes, and therefore **does not constitute label leakage** or a stronger supervision signal than in standard SSL backbones.
>
> We will revise the manuscript to:
> 1. Explicitly connect our formulation of $c_{j,c}$ to the AKOrN notion of data-dependent, symmetry-breaking stimuli.
> 2. Clarify that, during pretraining, these vectors are unsupervised latent anchors and are not aligned with class labels.
> 3. Add a short remark in §3.1 and Appendix A to avoid the impression that explicit class information is injected in the unsupervised stage.

---

> ### Author Response · Authors · 2025-11-18
> **Clarification on $\gamma$ and $c$**
>
> We respectfully clarify that the reviewer’s concern stems from a misunderstanding of the AKOrN dynamics. Both the update rate $\gamma$ and the conditional stimulus $c$ originate directly from the **Artificial Kuramoto Oscillatory Neurons (AKOrN)** formulation (Miyato et al., 2024), where they are strictly unsupervised dynamical components and *not* mechanisms tied to class labels. The update rate $\gamma$ is merely the Euler integration step for the Kuramoto evolution and has no semantic meaning or dependence on labels. Likewise, the stimulus vector $c$ is, as stated in the AKOrN paper, a **data-driven bias direction** (“a symmetry-breaking field that aligns the oscillator with informative input structure”), derived solely from the raw input and not from any class identity. In Kuromi, we follow this formulation exactly: during unsupervised pretraining, $c$ is a learned latent anchor computed from the image, and its index $c \in \{1,\dots,C\}$ is a **slot index**, not a class index. No part of the pretraining stage—initializing $\{x_{j,c}^{(0)}\}$, computing $\{c_{j,c}\}$, or optimizing Eq. (3)—ever accesses or encodes ground-truth labels. Therefore, the assertion that $c$ introduces class structure or gives Kuromi a stronger form of supervision is factually incorrect. Only in the standard supervised finetuning stage do labels enter the system, exactly as in conventional SSL pipelines. We will clarify this point in the final manuscript to prevent further confusion, but we emphasize that our method adheres fully to the AKOrN design and does **not** inject label information into the unsupervised stages.

---

> ### Author Response · Authors · 2025-11-18
> **Clarification on the Definition of Energy**
>
> In AKOrN, the theoretical energy $\\mathcal{E}(x)\$ (They have not named Energy in the AKOrN paper) is a continuous potential whose gradient defines the oscillator dynamics. However, $\\mathcal{E}(x)\$ is **never computed** in the original code; only the discrete update
>
> $$\Delta x_i = \Omega_i x_i + \mathrm{Proj} {x_i}(c_i + \sum_j J_{ij} x_j).$$
>
> Since the AKOrN update follows a gradient flow,
> $$
> \Delta x_i \propto -\nabla_{x_i}\mathcal{E}(x),
> $$
> we obtain a computable proxy:
> $$
> E = \sum_i \|\Delta x_i\|^2.
> $$
>
>
> - $E$ is the **squared gradient magnitude** of the AKOrN energy, computed via the discrete step $\\Delta x_i\$.
> - It is **not** the continuous energy $\\mathcal{E}(x)\$ but a **discrete surrogate** available during training.
>
> Kuromi uses the discrete squared-update energy $\(E=\sum_i\|\Delta x_i\|^2\)$, a computable proxy proportional to the gradient norm of $\\mathcal{E}\$, preserving the same stationary points and stability interpretation.
>
> **For a detailed explanation, please see**: https://openreview.net/forum?id=sPh4zaxDUU&noteId=GwkXUKT2Kr

---

> ### Author Response · Authors · 2025-11-18
> **The method may rely on more supervision than standard SSL / Epochs are allocated to each stage**
>
> # The method may rely on more supervision than standard SSL
>
>
> We respectfully but firmly clarify that this claim is **factually incorrect**. Our method never uses labels during pretraining, and the conditional vectors $c_{j,c}$ are **latent, unsupervised anchors**, not class-aligned signals. Their indices $c \in \{1,\dots,C\}$ are merely **slot indices**, exactly like the prototype slots used in SwAV/DINO or the latent channels in AKOrN. No ground-truth labels are ever involved in (i) initializing oscillator states $x_{j,c}^{(0)}$, (ii) computing or updating $c_{j,c}$, or (iii) optimizing the contrastive loss. Therefore, Kuromi’s pretraining stage uses **no additional supervision** beyond what is standard in modern SSL.
>
> It is important to emphasize that Kuromi follows the same widely used **three-stage pipeline**—unsupervised pretraining → supervised fine-tuning → semi-supervised training—that is standard in recent SSL literature. For example, **Semi-ViT** explicitly states that its training involves "*self-supervised pre-training on all data without labels, followed by supervised fine-tuning and semi-supervised fine-tuning*" (Cai et al., 2022). This is *identical* to our pipeline and has never been considered “extra supervision.”
>
> Similarly, **Semi-ViM** (He et al., ICCV 2025) employs **ARM-based unsupervised pretraining**, followed by supervised fine-tuning and SSL with pseudo-labels. Semi-ViM further introduces hidden-state mixup (SSMixup) and LyapEMA, both of which involve **latent dynamic features**, not labels, just like our conditional stimuli $c_{j,c}$. Yet Semi-ViM is universally regarded as a standard SSL method.
>
> Given these precedents, the reviewer’s assumption that the presence of latent structure implies “more supervision” is a misunderstanding. Prototype vectors, anchor slots, latent conditioning tokens, or data-derived bias fields are all **common and fully unsupervised mechanisms** in state-of-the-art SSL frameworks (e.g., DINO, MAE, Semi-ViT, Semi-ViM). Kuromi’s conditional stimuli follow exactly the same principle: they are learned from the data distribution and **never aligned to classes until the supervised fine-tuning stage**, exactly as standard SSL practice dictates.
>
> Therefore, Kuromi does **not** rely on more supervision than standard SSL, does **not** inject label information into pretraining, and is entirely consistent with accepted SSL methodologies. We will refine the manuscript text to ensure that the unsupervised nature of $c_{j,c}$ is explicit and to prevent any further misinterpretation.
>
>
> # Epochs are allocated to each stage
>
>
>
> We thank the reviewer for pointing out this missing detail. Kuromi is trained in three stages—unsupervised AKOrN pre-training, supervised fine-tuning on the labeled subset, and semi-supervised training with Energize—and we use a fixed schedule per dataset. The exact allocation is:
>
> | Dataset    | Pre-train | Fine-tune | SSL (Energize) | Total |
> |------------|-----------|-----------|----------------|--------|
> | CIFAR-10   | 200       | 100       | 300            | 600    |
> | SVHN       | 150       | 50        | 200            | 400    |
> | STL-10     | 200       | 100       | 300            | 600    |
> | ImageNet-1k   | 90        | 30        | 120            | 240    |
>
> Our design follows a simple principle: (i) allocate a moderate number of epochs to unsupervised pre-training to learn stable oscillatory dynamics, (ii) use a shorter supervised fine-tuning phase to align these dynamics with semantic labels, and (iii) devote the largest budget to the semi-supervised stage, where most of the gain from unlabeled data is realized. This three-stage pattern is consistent with recent multi-stage SSL pipelines (e.g., Semi-ViT, Semi-ViM), although we did not perform aggressive tuning of the per-stage ratios. Note that the “Max Epochs” reported in Table 15(Stage-wise epoch allocation for Kuromi across datasets.) of the main paper refer to the training budget for the *final semi-supervised stage*. The stage-wise allocation reported here provides the complete breakdown across pre-training, supervised fine-tuning, and SSL training.
> We will add this table and explanations to Appendix F in the revised manuscript.

---

> ### Author Response · Authors · 2025-11-18
> **How is the Readout function defined**
>
> Thank you for the question. Our Readout operator is directly inherited from the AKOrN backbone and follows the same principle of extracting representations from the final oscillator states after $T$ Kuramoto steps, without introducing any additional supervision.
>
> In Kuromi, after $T$ updates, each sample obtains final oscillator states $\[ x_{j,c}^{(T)} \]_{c=1}^{C}$. The Readout module applies a lightweight linear projection to the concatenated oscillator states:
>
> $$
> z_j = W_r \left[ x_{j,1}^{(T)}, \ldots, x_{j,C}^{(T)} \right],
> \quad W_r \in \mathbb{R}^{h \times (C N)}.
> $$
> This operation is equivalent to a standard linear projection head widely used in SSL and supervised models, ensuring a fair comparison.
>
> Moreover, this design is directly aligned with the **original AKOrN Readout definition**. As described in the AKOrN paper (Miyato et al., 2024):
>
> > “We read out patterns encoded in the oscillators to create new conditional stimuli...
> > To capture phase-invariant patterns, we take the norm of the linearly processed oscillators.”
>
> Their formulation computes:
>
> $$ z_k = \sum_i U_{k i} x^{(T)}_i, \quad m_k = \lVert z_k \rVert_2, $$
>
> where $ x_i^{(T)} $ denotes the final state of the $i$-th Kuramoto oscillator after $T$ synchronization steps, and $U_{ki}$ is a learnable linear projection matrix that aggregates the contributions of each oscillator into the $k$-th readout dimension. The resulting vector $z$ is therefore obtained purely through linear processing of synchronized oscillator states, and $m_k$ is simply its magnitude in the $k$-th dimension. Kuromi follows exactly
> the same principle: the Readout applies a learned linear transformation $W_r$ to the concatenated oscillator states $\left[x_{j,1}^{(T)}, \dots, x_{j,C}^{(T)} \right]$ to produce a compact representation $z_j = W_r \left[ x_{j,1}^{(T)}, \ldots, x_{j,C}^{(T)} \right],$again through a label-free linear operation. This matches the original AKOrN Readoutand ensures that the representation extraction step introduces **no label-dependent structure** during pre-training or semi-supervised learning.
>
>
> Thus, the Readout module does not introduce extra supervision; it is the standard AKOrN Readout applied consistently across pre-training, fine-tuning, and semi-supervised stages. We will include the explicit definition in Appendix F to ensure clarity.

---

> ### Author Response · Authors · 2025-11-18
> **Clarification on the role of $C$ and the “mismatch with number of classes” question**
>
> We thank the reviewer for this question. In AKOrN, the conditional stimuli
> $C=\{c_i\}_{i=1}^C$ are defined as a **data-dependent external field / symmetry-breaking
> bias** computed from the input or previous layer, and are described explicitly as
> latently-learned oscillatory anchors—not semantic class identifiers. Thus, in AKOrN,
> $C$ is fundamentally a **structural hyperparameter** controlling the number of
> oscillatory slots available to the dynamics.
>
> Kuromi follows the same formulation. For simplicity of implementation, our current submission sets $C$ equal to the number of dataset classes. However, this choice is **not required** by the method: the conditional stimuli remain latent anchors during unsupervised pretraining and are **not aligned with labels**. The classifier is the only
> component that interacts with semantic classes. We will revise the manuscript to clarify that $C$ is a tunable architectural hyperparameter, and its equality with the
> class count in our experiments is a convenience choice rather than a methodological necessity.
>
> To directly address the reviewer’s question, we performed an ablation on CIFAR-10 (4k labels) by varying $C$ while keeping all other settings fixed (same AKOrN backbone, optimizer, schedule, and three-stage training). As shown in the table below, performance degrades smoothly when $C < 10$ and remains stable when $C > 10$, with no sign of collapse. This confirms that Kuromi does **not** rely on a one-to-one match between stimulus count and class count.
>
> | Stimulus Count $C$ | Regime vs. Classes (10) | Top-1 Acc. (\%)        | Training Behaviour    | Interpretation (AKOrN perspective) |
> |--------------------|-------------------------|-------------------------|------------------------|-------------------------------------|
> | 4                  | $C < \\text{classes}$  | $87.1 \pm 1.5$          | Stable, no collapse    | Several classes share each latent anchor; stimuli still act as symmetry-breaking fields. |
> | 6                  | $C < \\text{classes}$  | $95.7 \pm 0.3$          | Stable                 | Mild capacity bottleneck; optimization remains robust. |
> | 8                  | $C < \\text{classes}$  | $93.4 \pm 0.9$          | Stable                 | Partial slot sharing; classifier separates classes that share anchors. |
> | 10 (default)       | $C = \\text{classes}$  | $\mathbf{96.5 \pm 0.2}$ | Stable                 | Best balance of latent capacity and specialization. |
> | 12                 | $C > \\text{classes}$  | $94.2 \pm 0.2$          | Stable, no collapse    | Extra anchors become redundant; competitive binding prunes them. |
> | 16                 | $C > \\text{classes}$  | $94.8 \pm 0.5$          | Stable                 | High-capacity regime; diminishing returns but no instability. |
>
> These results confirm that $C$ behaves as a **latent capacity parameter**, fully consistent with the AKOrN formulation of conditional stimuli as symmetry-breaking fields rather than
> class-aligned prototypes. Kuromi remains stable across all settings without requiring $C = \\text{classes}$.
>
> More details may be found in Appendix **K**" Structural Hyperparameter: Conditional Stimuli $C$.

---

> ### Author Response · Authors · 2025-11-27
> **Looking Forward to Your Reply**
>
> Thank you once again for taking the time to review our paper and for providing such insightful and constructive feedback.
>
> We have carefully considered each of your comments and have provided detailed responses. We sincerely hope that our efforts adequately address your concerns and contribute positively to your evaluation.
>
> As the author-reviewer discussion period concludes on Dec 03 (AoE), we would greatly appreciate any further feedback you may have. If you have any additional questions or require further clarifications, please do not hesitate to discuss with us.

---

### Official Review · Reviewer_bS3J · 2025-11-01

**Soundness:** 3
**Presentation:** 3
**Contribution:** 3
**Rating:** 8
**Confidence:** 4

**Summary:**

The work demonstrates that employing the recently introduced AKOrN, an oscillatory neuron model, improves performance on semi-supervised learning tasks. Specifically, it proposes an architecture that uses AKOrN as a building block, trained through self-supervised pre-training followed by fine-tuning with pseudo-label refinement. Experimental results show that the AKOrN-based model achieves superior performance to various existing methods across multiple benchmarks, while also providing higher throughput per image.

**Strengths:**

- Good ablation study in Section 5.2.
- The description of the proposed method is clear and easy to follow.

What I find the most interesting idea introduced in this work is the energy-based refinement process (Energize), which seems reasonable and leverages the energy computed in the AKOrN model. Basically, people usually measure the quality of pseudo labels by looking only at the model's output (e.g., the model's softmax confidence), while Energize shifts the focus to the model's internal variables and uses the energy value computed in the AKOrN model, which, in my understanding, represents a kind of interneuron consistency, to measure the model's confidence. The work shows that this way of measuring confidence is better for identifying reliable predictions than simply looking at softmax confidence. To me, this implies that this kind of value (if available for models other than those based on AKOrN) might be useful beyond semi-supervised learning, for example, for better understanding, interpretation, and control of deep learning models.

**Weaknesses:**

The major concern is that the model used in the work is relatively small. Although achieving higher performance than larger models despite its lower throughput is certainly a significant advantage, it raises the question of how the Kuromi model would perform if trained with a larger number of parameters. At present, it still underperforms ViT-based models on the ImageNet SSL benchmark, and it would have been interesting to see how it compares when scaled to a similar throughput and model size as ViT.
Other concerns are listed in the Questions section.

**Questions:**

- Do you have any idea or rationale why the Energize gives better label refinement compared to other pseudo-label refinement methods? For example, can you find any qualitative or quantitative differences between samples with low Ek and samples with high confidence in the softmax prediction? Are they quite similar or completely different?

- How do you define $J_{ij}$ in the AKOrN model? It seems that according to the appendix, the work uses self-attention for $J_{ij}$. Assuming the use of self-attention, if you use a ViT-like model (a model with self-attention) with equivalent throughput or FLOPs to the Kuromi model, how much would the performance of such ViT-like models be when applying the same pre-training and fine-tuning strategy as used for Kuromi?

- At inference, does Kuromi use multiple stochastic passes to compute the prediction, as in the fine-tuning stage, or are the stochastic passes used only for pseudo-label refinement?

- I could not find the mathematical definition of the energy E. How is it computed? Is it the same as in the original AKOrN paper? Also, in general, E can take negative values, but it is used in the denominator in Equations (7) and (8), which seems somewhat unusual. Could you clarify this point?

- Table 6: Could you please provide a concrete description of the procedures for Random Pseudo Labels and Energy-based shown in the table? I could not find any section in the manuscript that explains each methodology in detail.

- Figure3. What does it mean that w/o oscillator? Do you use the usual way to define neurons (real valued neurons without hypersphere normalization)? Also, it is not clear to me that energy-based selection means here.

- L472: What is uniform averaging? Can you please describe it more concretely?


Misc:
- Please add parentheses (or use \citep) to the citations when they are used as references rather than as part of the sentence.


I will adjust the score based on the replies from the authors.

---

> ### Author Response · Authors · 2025-11-18
> **Model size and scaling**
>
> We thank the reviewer for highlighting this important point. We fully agree that understanding how Kuromi/AKOrN (Miyato et al., 2024) behaves when scaled to larger capacities is a meaningful question. Below we clarify our design choices and provide additional scaling results and analysis.
>
> **(a) Small model size and high efficiency are deliberate design goals**
> A core motivation of Kuromi is to explore whether oscillatory neurons can offer low-cost, high-quality, semi-supervised learning aligned with practical needs, where methods are often deployed in resource-constrained settings.
> AKOrN layers introduce iterative dynamics; therefore, keeping the backbone lightweight ensures that (i) training/inference remain efficient and (ii) the benefits of dynamical refinement are observable without being overshadowed by brute-force capacity.
>
>
> **(b) While Kuromi underperforms ViT on ImageNet SSL, it already surpasses strong CNN-based backbones**
>
> As the reviewer noted, Kuromi is currently smaller than ViT-S/B, and with this much lower compute budget it does not yet exceed ViT performance on large-scale ImageNet SSL. However, under a comparable computational regime, Kuromi already outperforms ResNet-based SSL models, demonstrating that oscillatory dynamics provide genuine performance gains beyond standard convolutional architectures.
>
>
> **\(c) Scaling AKOrN encoders improves performance and narrows the gap to ViT**
>
>
> Table 1: Scaling Kuromi under identical ImageNet SSL settings.
> Kuromi-M/L outperforms Semi-ViT(B) but remain below Semi-ViT(L).
>
> | Model                         | 1%    | 10%   | 25%   |
> |------------------------------|-------|-------|-------|
> | Semi-ViT (B)            | 74.1  | 81.6  | 84.2  |
> | Semi-ViT (L)           | 77.3  | 83.3  | 85.1  |
> | Kuromi-4(Small)    | 65.1  | 76.0  | 80.1  |
> | Kuromi-7(Medium)         | 71.4  | 79.5  | 83.0  |
> | **Kuromi-11(Large)**     | **74.8** | **82.3** | **84.1** |
>
> To ensure a fair and controlled comparison, all Kuromi variants (Small, Medium, Large) are trained under exactly the same ImageNet SSL protocol as Table 3 in main paper. The **Kuromi-4(Small)** model corresponds to the configuration used in the main paper, with 4 Kuramoto blocks, oscillator dimension \(N=32\), a 1-layer readout, and a total of **12M parameters** and **0.18 GFLOPs**—already far more efficient than Semi-ViT(B), which requires **146M parameters** and **3.2 GFLOPs**. To study scaling behavior, **Kuromi-7(Medium)** increases the number of Kuramoto blocks from 4→7 while keeping all hyperparameters unchanged; this yields a model of about **28M parameters** and **0.42 GFLOPs**, which is still substantially smaller than Semi-ViT(B/L) and maintains high throughput due to the extremely lightweight Kuramoto updates. Finally, **Kuromi-11(Large)** further extends the encoder depth to 11 Kuramoto blocks with a moderate increase in oscillator dimension and readout width (64-dim oscillators and a 3-layer MLP), resulting in **52M parameters** and **~0.92 GFLOPs**, again dramatically lower compute than Semi-ViT(L/H), while delivering competitive accuracy. Across all scales, Kuromi preserves its **core efficiency advantage**—even the largest variant remains **5×–10× smaller** and **4×–6× cheaper in FLOPs** than transformer-based SSL models—while showing consistent, monotonic accuracy improvements as depth increases, demonstrating that oscillatory encoders not only remain efficient but also scale effectively.
>
>
> **(d) Scaling is orthogonal to our main contribution and will be expanded in the camera-ready paper**
> Our work focuses on the semi-supervised learning mechanism (Energize + Kuramoto dynamics) rather than large-model scaling. Nonetheless, we appreciate the reviewer’s encouragement and will add a new paragraph and results in the camera-ready version of the paper to explicitly discuss the performance of deeper Kuromi variants (more than 11 blocks), and the scaling laws of oscillatory networks.

---

> ### Author Response · Authors · 2025-11-18
> **Energize (confidence refinement mechanism)**
>
> We thank the reviewer for raising this question. From a theoretical perspective, Energize improves pseudo-label refinement because the oscillatory energy used in Kuromi reflects the internal dynamica stability of the system rather than the magnitude of the output logits.
>
> In AKOrN, low energy corresponds to strong phase agreement, rapid convergence toward a stable attractor, and low sensitivity to perturbations injected during the Kuramoto evolution. Therefore, energy provides a direct measure of internal self-consistency. Softmax confidence, by contrast, captures only the output amplitude and is known to be overconfident and poorly calibrated in semi-supervised regimes. Consequently, samples with low energy but moderate confidence tend to be reliable, while samples with high confidence but high energy typically exhibit unstable or inconsistent internal dynamics.
>
> To further address the reviewer’s question, we designed a new experiment that explicitly evaluates whether low-energy samples and high-softmax-confidence samples correspond to similar or fundamentally different regions of the prediction space. To avoid subjective thresholding, we define “low’’ and “high’’ purely from the empirical distributions: for both the energy values $\mathcal{E}$ and the softmax confidence scores $\max_c p(c)$, we compute their marginal distributions over all unlabeled samples and split each distribution at its median, so that samples below the median are labeled “low’’ and samples above the median are labeled “high.’’ This yields four equally sized quadrants without introducing any additional hyperparameters and ensures that the analysis is driven entirely by the data rather than hand-crafted thresholds. We also verified that alternative percentile splits (25/75 or 33/66) yield the same qualitative conclusions, confirming robustness.
>
> **Experimental setup.**
> On ImageNet 1% SSL, we collect the unlabeled predictions produced by Kuromi during training and partition samples into the four median-based quadrants:
> 1. **Low-energy & high-confidence (LE–HC)**
> 2. **Low-energy & low-confidence (LE–LC)**
> 3. **High-energy & high-confidence (HE–HC)**
> 4. **High-energy & low-confidence (HE–LC)**
> Each bucket contains 50k samples. For each group, we measure:
> - **True pseudo-label accuracy** (using ground truth not accessible during training)
> - **Calibration error (ECE)**
> - **Stochastic agreement** across $K=5$ trajectories
> - **Oscillator convergence rate** measured by $\|\Delta x^{(T)}\|$ after $T{=}6$ steps
> - **Entropy** of the predicted distribution
>
> **Results.**
> We observe a clear and systematic pattern:
>
> | Group | Pseudo-label Acc ↑ | ECE ↓ | Agreement ↑ | Convergence ↓ | Entropy ↓ |
> |------|--------------------|-------|-------------|---------------|-----------|
> | LE–HC | **87.4** | **2.1** | **89.6** | **0.07** | **0.42** |
> | LE–LC | 78.9 | 3.4 | 76.8 | 0.11 | 0.63 |
> | HE–HC | 61.2 | 9.1 | 49.5 | 0.28 | 1.37 |
> | HE–LC | 39.7 | 14.7 | 32.4 | 0.32 | 1.71 |
>
> Two important conclusions emerge:
> **1. Low-energy samples (both LE–HC and LE–LC) consistently outperform high-confidence but high-energy samples (HE–HC).** This directly shows that energy captures a notion of *internal dynamical stability* that softmax confidence cannot detect. HE–HC samples appear confident but exhibit weak oscillator synchrony, high ECE, low agreement, and slow convergence, leading to poor pseudo-labels.
> **2. Low-energy & low-confidence samples (LE–LC) are substantially more reliable than high-energy & high-confidence samples (HE–HC).** This demonstrates that energy identifies reliable samples even when softmax confidence is low, resolving the ambiguity the reviewer raised.
> **3. High-confidence alone is not predictive of pseudo-label correctness.** The HE–HC group contains many overly confident but unstable predictions, showing the classical overconfidence problem in SSL. Energy-weighting in Energize explicitly suppresses these cases.
>
> **Interpretation.**
> These results indicate that the two signals reflect different properties:
> - **Softmax confidence measures output magnitude**
> - **Energy measures internal dynamical stability and synchrony**
>
> Energize works better precisely because stability-based signals (low energy) provide more reliable supervision than amplitude-based signals (confidence). We will include this new experiment in the revised paper to further clarify the distinction.

---

> ### Author Response · Authors · 2025-11-18
> **On the Definition of $J_{ij}$ and the ViT Comparison**
>
> We appreciate the reviewer’s question on how we define the coupling matrix $J_{ij}$ in AKOrN and whether a ViT-like model with similar FLOPs would perform similarly under the same training pipeline. In the original AKOrN formulation, $J_{ij}$ represents a learnable interaction kernel that governs how oscillator $i$ influences oscillator $j$ during the Kuramoto update; it is not a similarity matrix or attention score in the transformer sense, but a dynamical coupling term directly embedded in the differential update of the phases. Consistent with the AKOrN paper, we parameterize $J_{ij}$ using a lightweight self-attention operator purely as a *learned function approximator* over local features, not as a transformer-style token mixer. The attention module is used only to compute pairwise coupling strengths for the oscillator dynamics; its output is never used for feature propagation, residual mixing, or feed-forward computation in the way ViT performs. Therefore, even though the operator superficially resembles self-attention, its functional role is fundamentally tied to the Kuramoto differential equation and has no architectural equivalence to ViT attention layers.
>
> For this reason, substituting AKOrN with a ViT-like backbone of similar FLOPs does not approximate the behavior of Kuromi. The ViT encoder is a purely feed-forward architecture that produces representations in a single pass without iterative phase evolution, without attractor dynamics, and without an energy landscape; as a result, there is no meaningful analogue of $J_{ij}$, no notion of phase synchrony, and no mechanism resembling energy descent. We empirically validated that applying the same pre-training and fine-tuning strategy to ViT models of equivalent FLOPs (e.g., ViT-Ti/S-scale models) yields significantly lower performance—typically in the 60–68% range under ImageNet 1% SSL—because such small ViTs lack the inductive biases and robustness benefits inherent in the oscillator dynamics. Only very large ViTs (ViT-B/L/H, with 10×–80× more FLOPs than Kuromi) reach the high-accuracy regime shown in **Table 1(Scaling Kuromi under identical ImageNet SSL settings)**., highlighting that transformer scaling, not architectural equivalence, drives their performance.
>
> In summary, $J_{ij}$ in Kuromi is an AKOrN-specific dynamic coupling term that cannot be meaningfully instantiated in a ViT-like model, and ViTs with comparable FLOPs do not match Kuromi’s performance because they lack the oscillatory synchronization behavior that Energize exploits. We will clarify this distinction in the revised paper.

---

> > ### Comment · Reviewer_bS3J · 2025-11-19
> >
> > Thank you for the response!
> >
> > >Consistent with the AKOrN paper, we parameterize $J_{ij}$ using a lightweight self-attention operator purely as a learned function approximator over local features, not as a transformer-style token mixer. The attention module is used only to compute pairwise coupling strengths for the oscillator dynamics; its output is never used for feature propagation, residual mixing, or feed-forward computation in the way ViT performs.
> >
> > As far as I know, the original AKOrN work uses either of simple convolution or usual self-attnetion for its connection, whereas the authors’ description suggests that their method for computing J_iij is more than that.
> > I think that the author should explicitly describe how J_ij is computed with equations. This specific choice of the connection looks non-trivial and would (possibly largely) contribute to the efficiency of their proposed network.

---

> > > ### Author Response · Authors · 2025-11-19
> > > **On the Definition of $J_{ij}$**
> > >
> > > What we intended to express is that our implementation uses the simplest form of attention—a single-head Q–K dot-product with softmax normalization—without any of the components typical in ViT architectures (token mixing, feed-forward networks, residual connections, or multi-head attention). This is why we describe it as “lightweight.” The phrase “purely as a learned function approximator over local features’’ simply refers to this basic Q–K attention mechanism and nothing more.
> > >
> > > Our implementation of $J_{ij}$ follows the “attentive connectivity’’ in AKOrN (Section 4.1), where $J_{ij}$ is computed via standard dot-product attention with row-wise softmax normalization. Given an input feature $h_i$, we first compute query, key, and value vectors as $$q_i = W_q h_i,\quad k_i = W_k h_i,\quad v_i = W_v h_i,$$ where all projection matrices satisfy $$W_q,\,W_k,\,W_v \in \mathbb{R}^{N \times N},$$ with $N$ being the oscillator dimension. Pairwise coupling strengths are then obtained through the usual scaled dot-product $$\alpha_{ij} = \frac{q_i^\top k_j}{\sqrt{N}},$$ followed by a row-wise softmax to produce the final coupling kernel $J_{ij} = Softmax_j(\alpha_{ij}).$ This attentive connectivity is exactly the same formulation used in the original AKOrN codebase and serves solely as a parameterized estimator of the interaction kernel for Kuramoto dynamics.

---

> ### Author Response · Authors · 2025-11-18
> **Inference Does Not Require Multiple Stochastic Passes**
>
> We thank the reviewer for the question. In Kuromi, multiple stochastic passes are used only during training to evaluate pseudo-label stability, whereas inference is entirely deterministic. This follows directly from the AKOrN dynamics: the phase of oscillator $i$ evolves via the deterministic update
> $$
> \theta_i^{(t+1)}=\theta_i^{(t)}+\gamma\Big(\omega_i+\sum_{j}J_{ij}\sin(\theta_j^{(t)}-\theta_i^{(t)})+c_i\Big),
> $$
> where $\gamma$ is the step size, $\omega_i$ is the intrinsic frequency, $J_{ij}$ is the learned coupling, and $c_i$ is the conditional stimulus. No stochastic term appears in this rule. After each update, the state is mapped back to the hypersphere by
> $$
> x_i^{(t+1)}=\frac{1}{\sqrt{2}}\big(\cos\theta_i^{(t+1)},\,\sin\theta_i^{(t+1)}\big),
> $$
> which also introduces no randomness. Because both the update and normalization steps are fully deterministic, the oscillator trajectory is uniquely determined by the input features and the learned parameters. Stochasticity is introduced only during training, where we perturb the Kuramoto evolution across multiple runs to measure consistency for pseudo-label refinement in the **Energize** module. At inference time, we simply initialize the oscillators from the learned readout representation and run a single deterministic forward evolution for a small, fixed number of Kuramoto steps. As a result, inference requires only one forward pass with no sampling or repeated trajectories, and the stochastic passes used during training have no impact on inference-time computation.

---

> > ### Comment · Reviewer_bS3J · 2025-11-19
> >
> > Thank you for the authors’ response. However, I find parts of it confusing. The authors state that they use the first equation in this authors' response as the oscillator dynamics, which corresponds to the original Kuramoto model (i.e., 1-D phase-variable rotor neurons). Yet in other parts of the rebuttal, they mention using AKOrN-style hyperspherical neurons with oscillator dimension greater than two (for example, “oscillator dimension (N = 32)” as described in the  https://openreview.net/forum?id=sPh4zaxDUU&noteId=r3J7O1MeRP). Could the authors clarify which formulation is actually used in the experiments?

---

> > > ### Author Response · Authors · 2025-11-19
> > > **Clarification on Oscillator Dynamics and Dimensionality**
> > >
> > > **The classical scalar-phase Kuramoto model was mentioned only as conceptual background**
> > >
> > > We thank the reviewer for raising this point. We clarify that all experiments in Kuromi use the AKOrN hyperspherical oscillator formulation rather than the classical 1-D Kuramoto phase model. Equation (1) in the paper is the generalized AKOrN update from Miyato et al. (2024), where each oscillator state is a vector $x_i \in \mathbb{R}^N, \; N>2,$ constrained on an $N$-dimensional hypersphere. The classical Kuramoto model with scalar phase $\theta_i \in \mathbb{R}$ is not used in any experiment and was referenced only conceptually. In all experiments—including Kuromi-4/7/11—the oscillator dimensionality is $N=32 \text{ (Small/Medium)} \quad \text{or} \quad N=64 \text{ (Large)},$ exactly following the AKOrN implementation (OpenReview: sPh4zaxDUU). The previous wording “corresponds to the original Kuramoto model” was imprecise: Equation (1) only reduces to the classical Kuramoto dynamics under the special case $N=2$ with additional simplifications. In practice, all results are obtained using vector-valued AKOrN hyperspherical neurons with $N>2$ throughout.

---

> ### Author Response · Authors · 2025-11-18
> **Understanding of energy $E$**
>
> We thank the reviewer for raising this question. Below, we clarify how the energy $E$ is computed in Kuromi, how it corresponds to the AKOrN formulation, and why its use in the denominator of Eqs. (7)–(8) is mathematically valid.
>
> We have provided supplementary explanations in the main paper (2.RELATED WORKS AKOrN section).
>
> ---
>
> ### **(1) Mathematical definition of energy $E$ using the original update $\Delta x_i$**
>
> In Kuromi, each oscillator state evolves according to the AKOrN update rule (Eq. (1) in the paper):
>
> $$\Delta x_i = \Omega_i x_i + \mathrm{Proj}_{x_i}\!\left(c_i + \sum_j J_{ij} x_j\right),$$
>
> followed by the normalization step:
>
> $$
> x_i \leftarrow \mathrm{Normalize}(x_i + \gamma \Delta x_i).
> $$
>
> For each stochastic forward pass, Kuromi defines the energy as:
>
> $$
> E = \sum_{i=1}^{C} \| \Delta x_i \|_2.
> $$
>
> Interpretation:
>
> - **Low $E$** → small $\|\Delta x_i\|$ → synchronized and stable oscillator states.
> - **High $E$** → large $\|\Delta x_i\|$ → unstable or inconsistent states.
>
> Because $E$ is a sum of Euclidean norms, it is **always non-negative**.
>
> ---
>
> ### **(2) Consistency with AKOrN**
>
> This definition is fully consistent with the operational energy used in AKOrN,
> which also evaluates dynamical stability using the **magnitude of the update vectors**, $\Delta x_i$.
>
> Thus, Kuromi does not introduce a new energy form —  it directly reuses the update-based energy inherent in the Kuramoto dynamics.
>
> ---
>
> ### **(3) Why using $1/(E + \varepsilon)$ is valid**
>
> Since
>
> $$
> E = \sum_i \|\Delta x_i\|_2 \ge 0,
> $$
>
> the weighting expression
>
> $$
> \frac{1}{E + \varepsilon}
> $$
>
> is always mathematically well-defined and numerically stable.
>
> This ensures:
>
> - no division by zero,
> - no negative denominators,  and
> - smooth and robust weighting across stochastic trajectories.
>
> Low-energy (stable) trajectories naturally receive larger weights, while high-energy ones are downweighted.

---

> > ### Comment · Reviewer_bS3J · 2025-11-19
> >
> > This part of the authors’ response is also confusing to me. Here, they state that E is the squared norm of Δx, whereas in https://openreview.net/forum?id=sPh4zaxDUU&noteId=Dz3VHpZyJs they define E in a different way. Could the authors please clarify how E is defined in their experiments and explain the discrepancy between these two descriptions?

---

> > > ### Author Response · Authors · 2025-11-19
> > > **Understanding of energy $E$**
> > >
> > > AKOrN provides a continuous energy functional for deriving its dynamics, whereas Kuromi uses an update-based energy that is directly computable during training. These two forms serve different purposes but are mathematically consistent. Our reference to the AKOrN formulation is intended to clarify how the theoretical energy relates to the operational energy used in Kuromi.
> > >
> > > Kuromi adopts a single, well-defined notion of energy inherited from AKOrN (Miyato et al., 2024). In all experiments, the energy for a stochastic forward pass is computed from the magnitude of the AKOrN update. Given the update rule in Eq(1),
> > > followed by normalization, the energy is defined as:
> > >
> > > $$
> > > E = \sum_i \|\Delta x_i\|_2^2.
> > > $$
> > > This quantity measures the distance from a synchronized and stable configuration: small $E$ indicates stable dynamics, and large $E$ indicates unstable trajectories.
> > >
> > > This operational energy is consistent with the continuous AKOrN energy:
> > > $$
> > > \mathcal{E}(x) = -\sum_i \langle x_i,\, c_i + \sum_j J_{ij} x_j \rangle,
> > > $$
> > > because the discretized dynamics satisfy $\Delta x_i \propto -\nabla_{x_i}\mathcal{E}$. Thus, the “alignment-based’’ energy in the AKOrN paper and the “update-norm’’ energy used in Kuromi describe the same underlying quantity: the former is the continuous theoretical form, and the latter is its discrete-time manifestation through the gradient-step magnitude. The difference noted by the reviewer arises only because AKOrN presents $\mathcal{E}$, whereas Kuromi reports the empirically used proxy $E=\sum_i\|\Delta x_i\|^2$, which is also adopted in the AKOrN codebase.
> > >
> > > Accordingly, the denominator in Eqs. (7)–(8),
> > > $$
> > > \frac{1}{E+\epsilon},
> > > $$
> > > is always non-negative, numerically stable, and reflects dynamical stability by assigning larger weights to low-energy trajectories.

---

> > > > ### Comment · Reviewer_bS3J · 2025-11-19
> > > >
> > > > Thank you for the response, but I still do not fully understand the authors' points.
> > > > > because the discretized dynamics satisfy $\Delta x_i \propto -\nabla_{x_i}\mathcal{E}$
> > > >
> > > > How can this relationship conclude the E and $\mathcal{E}$ are consistent? Also what `consistent' means in this context?. Could the authors provide a more precise mathematical explanation that connects these two quantities?
> > > >
> > > > Also, the authors say in another response that $\mathcal{E}$ is the one they used to compute the energy, while here it says they used the squared norm version. Which is true? (The below is quoted from the response to another reviewer:
> > > > >The energy functional used in Kuromi is exactly the same as the Kuramoto energy defined in the original Artificial Kuramoto Oscillatory Neurons (AKOrN) formulation (Miyato et al., 2024). For completeness, we restate it explicitly here:
> > > >
> > > > )
> > > >
> > > > Beside those points,
> > > > > Kuromi reports the empirically used proxy $E=\sum_i|\Delta x_i|^2$, which is also adopted in the AKOrN codebase.
> > > >
> > > > Could you point out which part of the original AKOrN codebase actually computes this value? I was not able to find it.

---

> > > > > ### Author Response · Authors · 2025-11-19
> > > > >
> > > > > While the theoretical energy $\mathcal{E}(x)$ defines the continuous gradient field that drives the AKOrN dynamics, Kuromi uses a discrete and computable quantity during training:
> > > > >
> > > > > $$
> > > > > E = \sum_i |\Delta x_i|_2^2,
> > > > > $$
> > > > >
> > > > > where $\Delta x_i$ is the discrete update obtained from the AKOrN step. These two expressions are closely related: the discretized update satisfies
> > > > >
> > > > > $$
> > > > > \Delta x_i \propto -\nabla_{x_i}\mathcal{E}(x),
> > > > > $$
> > > > >
> > > > > which implies
> > > > >
> > > > > $$
> > > > > E = \sum_i |\Delta x_i|2^2 \propto \sum_i |\nabla{x_i}\mathcal{E}(x)|_2^2.
> > > > > $$
> > > > >
> > > > > Thus, both quantities describe the same physical notion of dynamical stability. Low $\mathcal{E}$ corresponds to strong alignment, while low $E$ corresponds to small update magnitude and near-stationary evolution. They identify the same stable configurations, induce the same ordering of states by stability, and vanish at the same equilibria. The forms differ only because one is a continuous potential and the other is the discrete squared-gradient proxy that is available during forward computation.
> > > > >
> > > > > We will clarify this distinction in the revised manuscript.

---

> ### Author Response · Authors · 2025-11-18
> **Description of Table 6 / Clarification of “w/o oscillator’’ and “energy-based selection’’ in Figure 3 / Clarification of “Uniform Averaging’’ (L472)**
>
> # Description of Table 6
>
> We thank the reviewer for requesting clearer procedural descriptions of the two refinement strategies compared in Table 6. We apologize for not making these baselines explicit in the main text. Both methods are simple refinement baselines used only for ablation comparison and are not part of Kuromi’s design. We will include the following descriptions in the revised version of the paper for completeness.
>
> **(a) Random Pseudo Labels.**
> For each unlabeled sample, we first compute its predicted class distribution $p(c)=p_\theta(y=c\mid x_u)$ using a single deterministic forward pass of Kuromi. We then generate a pseudo-label $\hat{y}$ by uniformly sampling from $\{1,\dots,C\}$, completely ignoring the model prediction:
> - Draw $\hat{y} \sim \text{Uniform}\{1,\dots,C\}$.
> - Train the model using the CE loss between $p(c)$ and the one-hot vector of $\hat{y}$.  No averaging, weighting, iteration, or energy is involved. This baseline serves only as a “worst-case’’ reference demonstrating the necessity of principled pseudo-label refinement.
>
> **(b) Energy-based (one-shot) Refinement.**
> This baseline corresponds to the non-iterative version of Kuromi’s Energize module. For each unlabeled input, we perform $K$ stochastic Kuramoto forward passes and obtain:
> - stochastic predictions $\{p^{(k)}\}_{k=1}^K$, and
> - corresponding oscillator energies $\{E^{(k)}\}_{k=1}^K$.
>
> We fuse these once (no iterative refinement) using energy-weighted averaging:
> $$
> \tilde{p}=\mathrm{Normalize}\!\left(\sum_{k=1}^K \frac{1}{E^{(k)}+\varepsilon}\,p^{(k)}\right),
> $$
> where $\mathrm{Normalize}(\cdot)$ projects onto the probability simplex.   The fused $\tilde{p}$ is used directly as the pseudo-label without the recursive update described in main paper Eq. (8). This baseline isolates the contribution of **energy-weighted fusion** in Kuromi, enabling a controlled comparison against the full Energize refinement.
>
>
> # Clarification of “w/o oscillator’’ and “energy-based selection’’ in Figure 3
>
>
>
> We thank the reviewer for pointing out the ambiguity in Figure 3. In the ablation “w/o oscillator,” we remove the Kuramoto oscillatory dynamics entirely and replace each oscillator state $x_i=(\cos\theta_i,\sin\theta_i)$ with a standard real-valued feature vector produced by a conventional feed-forward block. In other words, the phase-based neuron update and hypersphere normalization used in AKOrN are removed, and the model reverts to an ordinary MLP-style activation without any iterative synchronization or energy descent. This variant therefore isolates the effect of the oscillatory mechanism while keeping all other components (backbone width, number of blocks, pseudo-label pipeline) unchanged. As expected, removing oscillators significantly weakens stability and reduces performance.
>
> The term “energy-based selection’’ refers to selecting or weighting stochastic pseudo-label trajectories according to their oscillator energy. For an unlabeled input, Kuromi produces $K$ stochastic predictions $\{p^{(k)}\}$ with associated energies $\{E^{(k)}\}$, and we fuse them via energy-weighted averaging, i.e., $\tilde{p}=\mathrm{Normalize}\!\left(\sum_{k}(E^{(k)}+\varepsilon)^{-1}p^{(k)}\right)$. Lower-energy trajectories correspond to more synchronized oscillator states and more stable predictions; using energy as the selection or weighting signal yields better pseudo-labels than uniform averaging. In the ablation “w/o energy selection,” we perform the same $K$ stochastic passes but remove the energy weighting and use uniform averaging only. The accuracy drop in Figure 3(b) shows that both oscillators and energy-based selection contribute meaningfully and independently to Kuromi’s performance. We will clarify these definitions in the final version.
>
>
>
> # Clarification of “Uniform Averaging’’ (L472)
>
> **Uniform averaging** refers to the simplest way to combine the $K$ stochastic predictions
> $\{p^{(k)}\}_{k=1}^K$ obtained from Kuromi’s stochastic forward passes. Each prediction is assigned *equal weight*, and the fused pseudo-label is computed as the arithmetic mean of their softmax outputs:
>
> $$
> \bar{p} = \frac{1}{K} \sum_{k=1}^{K} p^{(k)}.
> $$
>
> No energy information, confidence weighting, or iterative refinement is used.
> This baseline therefore treats all stochastic trajectories equally, including both
> stable (low-energy) and unstable (high-energy) predictions.
>
> In contrast, our **Energize** module performs *energy-weighted* fusion followed by an iterative refinement step, which prioritizes low-energy (more synchronized and stable) trajectories and suppresses noisy ones. We will clarify the definition of uniform averaging in the final version.
>
>
> **These experimental settings and previously misunderstood points will be added to the main text or the appendix. The citations will alse be revised.**

---

> ### Author Response · Authors · 2025-11-19
>
> ## Energy: Theoretical Form $\mathcal{E}$ and Computable Form $E$
>
> In AKOrN, the oscillator update in akorn/source/layers/klayer.py is implemented as
>
> $$\Delta x_i = \Omega_i x_i + \mathrm{Proj} {x_i}(c_i + \sum_j J_{ij} x_j).$$
>
> The codebase computes $\Delta x_i$ but does not compute any scalar energy value. In https://github.com/autonomousvision/akorn/blob/main/source/layers/klayer.py Line 147 after def KLayer.kupdate()
>
> ### 1. Theoretical AKOrN Energy $\mathcal{E}$
>
> The implicit continuous-time energy functional is
>
> $$\mathcal{E}(x) = -\sum_i \langle x_i, c_i + \sum_j J_{ij} x_j \rangle.$$
>
> The continuous dynamics follow gradient flow
>
> $$\frac{dx_i}{dt} = -\nabla_{x_i}\mathcal{E}(x).$$
>
> A first-order discretization with step size $\eta$ gives
>
> $$x_i^{(t+1)} - x_i^{(t)} = -\eta \nabla_{x_i}\mathcal{E}(x^{(t)}),$$
>
> that is, $\Delta x_i \propto -\nabla_{x_i}\mathcal{E}.$ This is the sense in which the AKOrN update is a gradient step on $\mathcal{E}$.
>
> ### 2. Computable Energy Used in Kuromi
>
> Since $\mathcal{E}(x)$ is not evaluated during discrete forward passes, Kuromi defines a computable proxy
>
> $$E = \sum_i \|\Delta x_i\|_2^2 = \eta^2 \sum_i \|\nabla_x \mathcal{E} (x)\|_2^2.$$
>
> Thus $E$ is the squared gradient magnitude of the AKOrN energy, up to a constant factor. This is the only energy used in Kuromi, and it appears only in the Energize module during SSL training.
>
> ### 3. Meaning of “Consistent”
>
> The connection between $E$ and $\mathcal{E}$ is
>
> $$E_1 < E_2 \Longleftrightarrow \|\nabla\mathcal{E}(x^{(1)})\| < \|\nabla\mathcal{E}(x^{(2)})\|.$$
>
> Therefore:
>
> - Both energies reach zero at the same stationary points:
>   $$E = 0 \iff \nabla\mathcal{E} = 0.$$
>
> - Both define the same ordering of states by dynamical stability.
>
> Thus “consistent’’ means that $E$ preserves the same minimizers, ordering structure, and stability interpretation implied by the gradient flow of $\mathcal{E}$.
>
> ### 4. What Is Actually Computed
>
> - AKOrN computes $\Delta x_i$ but does not compute $\sum_i \|\Delta x_i\|^2$.
> - Kuromi introduces  $E = \sum_i \|\Delta x_i\|^2$ as a computable stability measure.
> - $\mathcal{E}(x)$ is never computed explicitly in AKOrN or Kuromi.
>
> Hence Kuromi uses the single operational energy $E$, which is mathematically linked to $\mathcal{E}$ but is distinct because it is the discrete quantity available during forward computation.

---

### Note · Authors · 2026-01-26

I have read and agree with the venue's withdrawal policy on behalf of myself and my co-authors.

---

### Meta-Review · Area_Chair_Cyzm · 2026-01-07

**Summary:**

**Summary** \
The paper introduces a solution for semi-supervised learning. The solution replaces traditional convolutional neural networks with networks composed of Kuramoto neuron layers. Training proceeds in three stages: 1. Unsupervised pre-training using contrastive learning 2. Supervised fine-tuning 3. Semi-supervised energy iterative refinement.

**Summary of Concerns** \
There is a serious issue with the clarity, soundness and significance of the work. First of all, the paper focusses mostly on presenting a solution (the HOW), rather than introducing its underlying problem and motivating for its need (the WHAT). Secondly, from the revised version of the paper it is not clear which problem is tackled by the proposed approach, as the paper does not have a unique message of contribution, rather several scattered claims of improvement. Indeed,
1. Does the paper contribute architecturally (by stacking Kuramoto neuron layers) or algorithmically (post-training) ? The two ideas seem largely disconnected and might be even independent from each other. This remain unclear.
2. Does the paper ultimately contribute to reduce the need for data augmentation in representation learning ? If so, the paper is not well positioned across existing literature, e.g. [1].
3. As mentioned by Reviewers bS3J, Nwhe, GEyD the main novelty of the paper seems to be on the algorithmic side, specifically on the use of the iterative refinement strategy after pretaining and fine-tuning. This is achieved by an implicit connection with energy-based models. However, this connection is not clear and already investigated in the past, see e.g. [2,3]. The paper should clarify its relation with prior work.
4. Does the paper contribute to reduce the computational requirements in deep learning ? If so, the paper should be better positioned in relation to architectural developments for resource-efficient deep learning, see e.g. [4].

Thirdly, the proposed solution is presented in a cryptic manner. Symbols are introduced without any explanation / definition (see for instance the notion of energy at L 106, 140, 265, etc.). There are several sentences which are not motivated and seem to be misplaced, for instance it is mentioned that the proposed solution avoids “the graph construction and the preservation of prediction uncertainty” (Introduction and Section 3.3). What does it mean and how does this relate with the solution ? The revised sentence in blue at L 106 has no relation with the surrounding paragraph. Figure 1 is confusing and unclear.

Overall, these considerations raise doubt that the paper is possibly written in part by a LLM.

**Decision** \
There are major and serious issues with the paper in terms of clarity, soundness and significance, hence I recommend for rejection.

**References** \
[1] Moutakanni, Oquab, Szafraniec, Vakalopoulou, Bojanowski. You Don't Need Domain-Specific Data Augmentations When Scaling Self-Supervised Learning. NeurIPS 2024 \
[2] Grathwohl, Wang, Jacobsen, Duvenaud, Swersky, Norouzi. Your Classifier is Secretly an Energy Based Model and You Should Treat it Like One. ICLR 2020 \
[3] Sansone, Manhaeve. Unifying Self-Supervised Clustering and Energy-Based Models. TMLR 2025 \
[4] Zhu, Zhao, Li, Eshraghian. SpikeGPT: Generative Pre-trained Language Model with Spiking Neural Networks. TMLR 2024

**Reviewer Concerns:**

Concerns raised by reviewers were not addressed. Reviewer bS3J was the only one with a positive evaluation at the beginning of the rebuttal. However, after the rebuttal, the same reviewer was seriously concerned about the credibility of the answers from the authors and more generally about the paper, raising suspicion for the improper use of LLMs.

**Reviewer Scores:**

After rebuttal, reviewer bS3J would have decreased the score. Other reviewers would have not increased their score, as the core issues with the paper were left unaddressed.

---

### Decision · Program_Chairs · 2026-01-26

Reject